# Mismatch repair deficiency is not sufficient to elicit tumor immunogenicity

Peter M. K. Westcott [1,5] ✉, Francesc Muyas[2], Haley Hauck[1,6], Olivia C. Smith [1,6], Nathan J. Sacks[1], Zackery A. Ely[1,3], Alex M. Jaeger[1], William M. Rideout III[1], Daniel Zhang [1], Arjun Bhutkar[1], Mary C. Beytagh [1], David A. Canner[1], Grissel C. Jaramillo[1], Roderick T. Bronson[4], Santiago Naranjo[1], Abbey Jin[1], J. J. Patten [1], Amanda M. Cruz[1], Sean-Luc Shanahan[1], Isidro Cortes-Ciriano [2] ✉ & Tyler Jacks [1,4] ✉

DNA mismatch repair deficiency (MMRd) is associated with a high tumor mutational burden (TMB) and sensitivity to immune checkpoint blockade (ICB) therapy. Nevertheless, most MMRd tumors do not durably respond to ICB and critical questions remain about immunosurveillance and TMB in these tumors. In the present study, we developed autochthonous mouse models of MMRd lung and colon cancer. Surprisingly, these models did not display increased T cell infiltration or ICB response, which we showed to be the result of substantial intratumor heterogeneity of mutations. Furthermore, we found that immunosurveillance shapes the clonal architecture but not the overall burden of neoantigens, and T cell responses against subclonal neoantigens are blunted. Finally, we showed that clonal, but not subclonal, neoantigen burden predicts ICB response in clinical trials of MMRd gastric and colorectal cancer. These results provide important context for understanding immune evasion in cancers with a high TMB and have major implications for therapies aimed at increasing TMB.

Immunotherapy has revolutionized the treatment landscape of many cancers, particularly those with a high tumor mutational burden (TMB)[1–5]. Somatic mutations can generate neoantigens capable of eliciting tumor-specific T cell responses[6,7], and it is widely believed that increased TMB renders tumors susceptible to immune attack after ICB treatment. Indeed, multiple pan-cancer meta-analyses have shown that TMB is one of the strongest predictors of ICB response[8–10], leading to approval by the US Food and Drug Administration (FDA) of pembrolizumab (anti-PD-1) for all tumors based on high TMB alone. In particular, MMRd is associated with some of the highest TMBs observed in cancer (hypermutation)[11–15] and remarkable response rates to pembrolizumab[2,5]. However, more than half the patients with MMRd tumors do not durably respond and TMB does not stratify responders[2,5]. This

underscores a critical need to understand what factors beyond TMB mediate efficacy. There are also conflicting studies suggesting that TMB is an imperfect biomarker of immunotherapy response[16] and argue that FDA approval of pembrolizumab based on TMB alone may be too broad[17]. Specifically, TMB provides limited to no additional predictive value within specific subsets of cancer known to have high rates of response to ICB, like those with MMRd[2,5].

One factor that may dilute the power of TMB as a biomarker of ICB response is intratumor heterogeneity (ITH) of mutations or, simply defined, the fraction of mutations that are subclonal—present in a minority of tumor cells. Subclonal neoantigens could be deleted with minimal impact on tumor fitness[18] or fail to elicit productive T cell responses[19]. Indeed, it has been observed in human cancer that ITH is

[1]David H. Koch Institute for Integrative Cancer Research, Massachusetts Institute of Technology, Cambridge, MA, USA. [2]European Molecular Biology Laboratory, European Bioinformatics Institute, Hinxton, Cambridge, UK. [3]Department of Biology, Massachusetts Institute of Technology, Cambridge, MA, USA. [4]Rodent Histopathology Core, Harvard Medical School, Boston, MA, USA. [5]Present address: Cold Spring Harbor Laboratory, Cold Spring Harbor, NY, USA. [6]These authors contributed equally: Haley Hauck, Olivia C. Smith. ✉e-mail: westcott@cshl.edu; icortes@ebi.ac.uk; tjacks@mit.edu

associated with decreased T cell infiltration[20,21] and poor survival[22,23], whereas clonal neoantigens are predictive of response to ICB[23–25]. This concept has been exemplified in an experimental setting of ultraviolet light B (UVB)-induced hypermutation of melanoma cell lines[26]. It is reasonable to hypothesize that similar mechanisms are operating in MMRd cancers given their constitutive mutational instability. Although previous studies showed that MMRd mutagenesis in vitro renders cell lines immunogenic[27,28], it is unclear what the impact is of MMR loss in vivo in the presence of immunosurveillance, a process that exquisitely shapes tumor immunogenicity[6,29,30]. To address these questions, we developed autochthonous mouse models of sporadic MMRd lung and colon cancer via targeted ablation of key genes in the MMR complex, MutL homolog 1 (*Mlh1*) and MutS homologs 2, 3 and 6 (*Msh2*, *Msh3* and *Msh6*), and performed preclinical trials to determine ICB sensitivity.

## Modeling sporadic MMRd in cancer

We adapted the autochthonous mouse model of lung cancer developed in our laboratory[31] by breeding in a *Cas9*-expressing allele (*Kras^{LSL-G12D}*; *Trp53^{flox/flox}*; *R26^{LSL-Cas9}* (KPC)) or conditional *Msh2* knockout allele[32] (*Kras^{LSL-G12D}*; *Trp53^{flox/flox}*; *Msh2^{flox/flox}* (KPM)). Intratracheal delivery of lentivirus-expressing Cre and an *Msh2*-targeting single guide (sg)RNA (sgMsh2) into the former or alveolar type II, cell-specific, adenovirus-expressing Cre into the latter induced lung adenocarcinomas with efficient MSH2 knockout (Fig. 1a–c and Extended Data Fig. 1a–e). We also adapted an endoscope-guided submucosal injection technique[33] to deliver lentivirus with sgRNAs targeting the colon tumor suppressor, *Apc*, in tandem with *Msh2*, *Mlh1*, *Msh3* or *Msh6*, into the distal colon of mice with constitutive *Cas9* expression (Fig. 1b). This efficiently induced focal colon adenomas with *MMR* gene knockout (Fig. 1d and Extended Data Fig. 1f–i).

To confirm mutation of *MMR* genes and investigate the degree of TMB, we performed whole-exome sequencing (WES) on micro-dissected tumors at 16–20 weeks post-initiation, including 26 sgMsh2- and -3-control (sgCtl)-targeted KPC lung tumors, 15 and 6 SPC-Cre-targeted KPM and KP (*Msh2* wild-type) lung tumors and 5 sgMsh2-, 6 sgMlh1-, 2 sgMsh6-, 6 sgMsh3- and 5 sgCtl-targeted colon tumors (Supplementary Table 1). Targeted sequencing of *MMR* genes in these tumors confirmed a preponderance of frame-shifting insertions/deletions (indels) (Supplementary Table 1). KPM, sgMsh2-, sgMlh1- and sgMsh6-targeted lung and colon tumors showed increased burden of somatic single-nucleotide variants (SNVs), indels and microsatellite instability (MSI) scores (MSIsensor[34]), whereas sgMsh3-targeted colon tumors showed elevated levels of indels only (Fig. 1e,f and Extended Data Fig. 1j–m). Indels across all MMRd tumors were predominantly single nucleotide and enriched at homopolymer repeat microsatellites (Fig. 1f and Extended Data Fig. 1n). Mutational patterns were highly consistent with those observed in human MMRd colon cancer[35] (cosine similarity of 0.9 and 0.8), as determined by decomposition of mutational spectra into individual signatures from the Catalogue of Somatic Mutations in Cancer (COSMIC)[36,37] (Fig. 1g,h and Extended Data Fig. 1n–q). The MMRd signature (SBS-MMRd) comprised 92%, 78% and 75% of all SNVs in the mouse MMRd colon and lung tumor models and human MMRd colon cancer, respectively (Fig. 1f). It is interesting that one lung tumor had a Ser415Arg mutation in the exonuclease domain of DNA polymerase epsilon (*Pole*) and, consequently, a much higher TMB (Extended Data Fig. 1j).

We also performed WES on cell lines derived from sgMsh2-targeted lung tumors. Unexpectedly, these showed on average a tenfold greater TMB than sgMsh2-targeted lung tumors (Fig. 1e and Extended Data Fig. 1k), suggesting low tumor purity (unlikely, because *Msh2^{KO}* lung tumors were on average 72% pure), mutagenesis in culture or substantial ITH reduced by clonal selection on plastic. Arguing against extensive mutagenesis in vitro, WES of a single-cell clone revealed more than double the TMB of the parental line, despite a much smaller increase in TMB after 20 passages (Extended Data Fig. 1r). Consistent with high ITH, more sequencing reads supporting mutations private to the single

clone were found in the sequencing data of the parental compared with the unrelated control line (1,685 versus 451), albeit at variant allele frequencies (VAFs) below the threshold of mutation calling (Extended Data Fig. 1s,t), as previously observed in hypermutated gliomas[38].

Altogether, these results establish the utility of our models to recapitulate fundamental mutational processes underlying hypermutation in MMRd human cancer. The lower clonal TMB that we observed is probably due to neutral evolution in the absence of major selective bottlenecks, a feature of ablating MMR concomitantly with tumor initiation in our models. Although these models cannot capture the accumulation of mutations or clonal evolution over decades in human cancer, they are uniquely suited to study the role of ITH in immune dysfunction of cancers with high and low prevalence of MMRd alike. Importantly, ITH is associated with aggressive disease and decreased ICB response in humans[22–25], but it remains unclear how ITH impacts the immune response in MMRd cancers specifically.

## Sporadic MMRd in the mouse is not immunogenic

Next, we sought to determine the effects of sporadic MMRd on tumorigenesis in our models. Neither KPC nor KPM *Msh2^{KO}* models showed a significant difference in overall tumor burden or grade at either timepoint (Fig. 2a–d and Extended Data Fig. 2b,c). Notably, there was no difference in tumor infiltration by T cells (CD3+) in the sgMsh2-targeted model (Extended Data Fig. 2d,e) or infiltration by cytotoxic (CD8+), helper (CD4+) or regulatory (CD4+FOXP3+) T cells within tumors (Fig. 2e–g) or whole lungs (Extended Data Fig. 2g,h) in the KPM model at either timepoint. Similar to the lung, tumors in the colon model showed no difference in growth kinetics with *Msh2* targeting (Extended Data Fig. 2i).

To determine the sensitivity of these models to immunotherapy, we first performed preclinical trials with ICB (anti-CTLA-4/anti-PD-1) in the KPM model. We followed similar dosing as established in seminal preclinical studies that preceded the first human clinical trials of anti-CTLA-4 (ref. 39) and anti-PD-1 (ref. 40), although we continued treatment for longer (4 weeks). In addition, we included treatment with the chemotherapeutic combination of oxaliplatin and low-dose cyclophosphamide (Oxa/Cyc) alone and in combination with ICB (Fig. 2h–i), because Oxa/Cyc has been shown to synergize with ICB[41,42]. To our surprise, no significant differences were observed between KPM and KP mice across all treatment arms, in both longitudinal change (Fig. 2j) and final tumor burden at necropsy (Fig. 2k and Extended Data Fig. 2j). Consistent with a lack of immunogenicity, we observed no differences in tumor grade or burden at 16 weeks between KPM and KP mice treated continuously with CD4+ and CD8+ T cell-depleting antibodies (αCD4/8) (Extended Data Fig. 2k–p). These results were not unique to the lung, because ICB treatment failed to induce any responses in sgMsh2-targeted colon tumors (Fig. 2l,m and Extended Data Fig. 2q,r). Likewise, there was no significant difference in endpoint size of sgCtl- versus sgMsh2-targeted colon tumors after ICB or continuous αCD4/8 treatment (Fig. 2n). Altogether, these data demonstrate that MMRd in these models is not sufficient to increase immunogenicity or sensitivity of tumors to ICB, alone or in combination with chemotherapy, in stark contrast to previous reports in cell-line transplant models[27,28].

## MMRd drives mutational heterogeneity

To assess the clonal composition of mutations in our models, we estimated cancer cell fractions (CCFs)[43]. TMB was predominantly subclonal and most mutations were present in less than a quarter of cells in MMRd tumors from lung and colon (Fig. 3a,b and Extended Data Fig. 3a–c). These mutations also adhered perfectly to a theoretical model of neutral evolution of subclonal mutations in cancer[44] (Fig. 3c,d), consistent with the absence of selective events after tumor initiation in our models. To investigate ITH more deeply, we performed WES on eight single-cell clones derived from an sgMsh2-targeted lung tumor cell line (09-2). Importantly, before subcloning, we restored MMR by re-expressing *Msh2* on a bicistronic lentivirus conferring puromycin

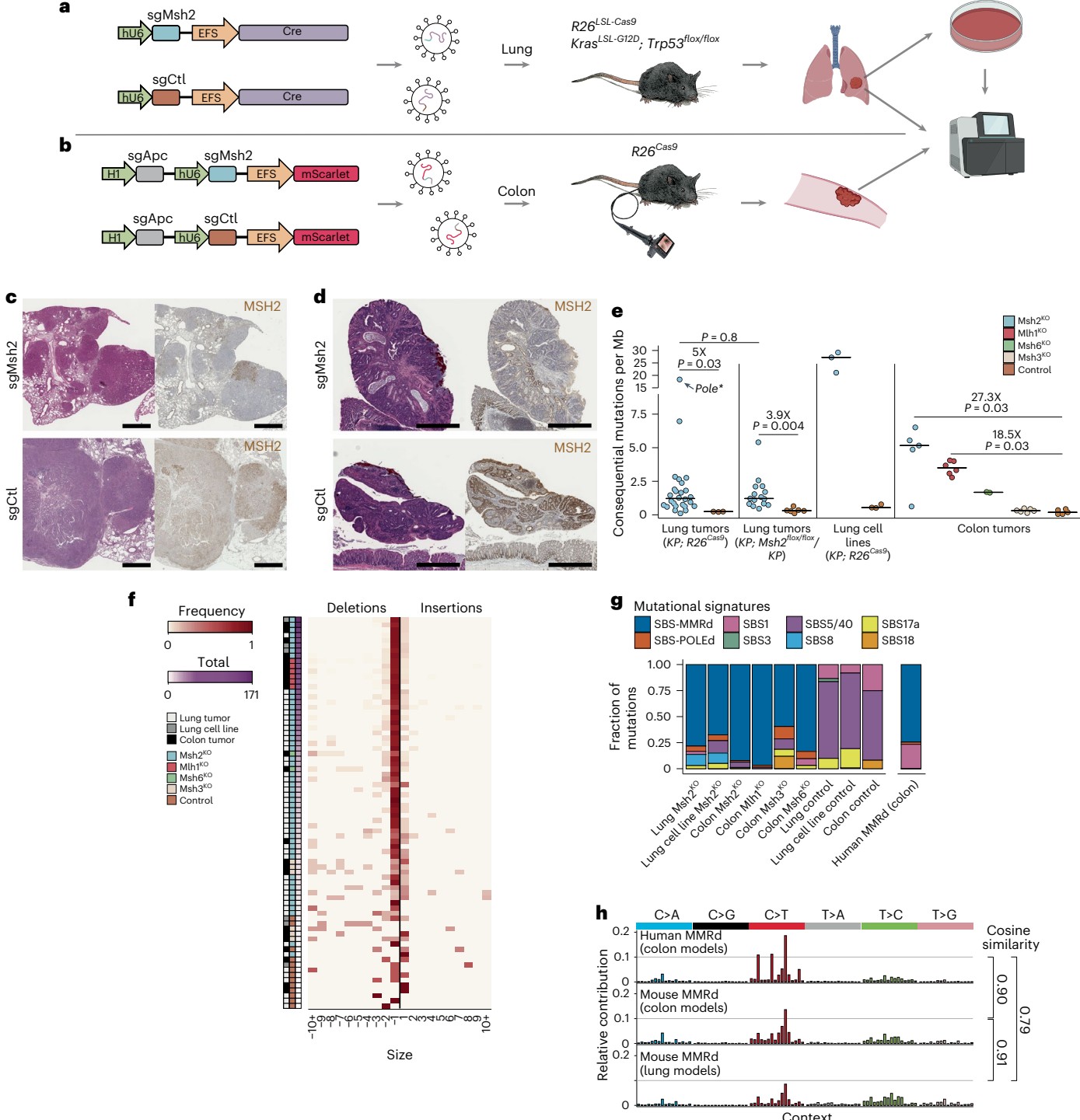

**Fig. 1 | Development of flexible in vivo models of DNA MMRd lung and colon cancer. a,b,** Schematic of lentiviral constructs and mouse strains used to induce MMRd lung (**a**) and colon (**b**) tumors for WES and in vitro analyses. **c,d,** H&E-stained and MSH2 IHC of sgMsh2- (top) and sgCtl-targeted (bottom) lung (**c**) and colon (**d**) tumors 16 weeks post-initiation, representing ten animals each. Scale bar, 1 mm. **e,** Total consequential mutations: nonsynonymous SNVs and indels per Mb of DNA for autochthonous lung tumors and cell lines and autochthonous colon tumors, with fold-change shown for each comparison. **f,** Frequency of indels from −10 nt to 10 nt across all sequenced autochthonous

tumors and parental cell lines, including exonic and intronic mutations. Samples were ordered by total indels. **g,h,** COSMIC mutational signature analysis of human MMRd colon and the mouse MMRd colon and lung tumors (**g**) based on frequencies of the 96 possible SNVs classified by substitution and flanking 5′- and 3′-bases, with cosine similarity score (**h**). Lung *Msh2^KO^*, *Kras^LSL-G12D^; Trp53^flox/flox^* (*KP*); *Msh2^floxl/flox^* and *KP* sgMsh2-targeted models combined. SBS_MRD, mismatch repair deficiency signature. Significance in **e** was assessed using Wilcoxon's rank-sum test with Holm's correction for multiple comparisons.

resistance (Fig. 3e). Of note, these clones (M1–8) maintained MSH2 expression and showed stable mutational and clonal architecture after 20 passages in the presence of puromycin (Extended Data Fig. 3d–f).

Nevertheless, significantly more somatic mutations were called in all clones than in the parental line (Fig. 3f) and these mutations were not broadly shared across clones (Extended Data Fig. 3g), supporting

the notion that ITH is underestimated by bulk sequencing methods[38]. Phylogenetic analysis of clones further confirmed a considerable level of ITH (Fig. 3g).

Given that ITH in our models arose in vivo, we sought to understand the role of immunoediting[6,29,30] in shaping this process. We performed WES on micro-dissected $Msh2^{KO}$ lung tumors (weeks 16–20) from animals continuously depleted of T cells ($n = 34$) or treated with ICB ($n = 12$). Continuous T cell depletion had no significant effect on overall TMB or tumor neoantigen burden (TNB), consistent with its lack of effect on tumor progression (Extended Data Fig. 2k–p). However, stratifying mutations by CCF revealed a significant increase in clonal (CCF ≥ 0.75), but not subclonal (CCF ≤ 0.5), TMB and TNB (Fig. 3h,i and Extended Data Fig. 3h,i). T cell-depleted tumors also showed a striking difference in CCF distribution and median CCF of expressed neoantigens, with significant enrichment of clonal neoantigens (Fig. 3j,k and Extended Data Fig. 3j). We observed similar results in the colon $Msh2^{KO}$ model (Extended Data Fig. 3k–n). It is interesting that tumors from ICB-treated animals showed significantly lower subclonal, but not clonal, TMB and TNB, suggesting possible simultaneous loss of subclonal populations and selective expansion of others in this context of ICB nonresponse (Fig. 3h,i and Extended Data Fig. 3h,i). Altogether, these results argue that neoantigens with high, but not low, clonal fraction are negatively selected by the adaptive immune system during tumor progression. By extension, immunosurveillance in mutationally unstable tumors promotes ITH by selectively pruning clonal neoantigens and thereby increasing the relative fraction of subclonal neoantigens. ICB treatment may lower the CCF threshold for T cell-mediated elimination, but probably also drives expansion of less immunogenic subclones that maintain an overall state of ITH in nonresponders.

## Intratumor heterogeneity enables immune evasion

To evaluate the impact of ITH, specifically tumor cell clonality, on immunogenicity in our models, we assessed survival of animals after orthotopic transplantation of the lung tumor cell lines and clones described above. The parental MSH2 knockout lines and low TMB control line were similarly nonimmunogenic, showing no difference in disease progression with and without continuous T cell depletion (Fig. 4a). This is consistent with the lack of immunogenicity of tumors induced by chemical carcinogens in immunocompetent animals and the principle of immunoediting[6,29]. Unlike MMRd in the autochthonous model, however, mice transplanted with the parental $Msh2^{KO}$ line (09-2) or an equal mixture of 09-2-derived $Msh2^{KO}$ clones (M1–8) and treated with ICB showed 20–30% durable responses and reduced hazard ratios (HRs) (0.34–0.50) over 30 weeks (Fig. 4b). Mice transplanted individually with five otherwise nonimmunogenic $Msh2^{KO}$ clones (M1, M2, M4, M5 and M6) and treated with ICB showed 30–75% durable responses and even further reduction in HRs (0.05–0.32) over 30 weeks (Fig. 4b). It is

interesting that the other three $Msh2^{KO}$ clones (M3, M7 and M8) were strongly immunogenic even without ICB treatment, efficiently driving disease only with T cell depletion (Fig. 4b). These differences were not the result of tumor intrinsic growth rates or loss of antigen presentation, because all clones showed similar in vitro growth kinetics and readily expressed major histocompatibility complex class I (MHC-I) (H-2K^b, H-2D^b) and programmed death-ligand 1 (PD-L1) on stimulation with interferon-γ (IFN-γ; Extended Data Fig. 4a–e). Although M3, M7 and M8 did not show increased TMB/TNB relative to the other clones, they did express on average higher H-2K^b/H-2D^b with IFN-γ stimulation, particularly the most immunogenic clone, M3, although this was not significant (Extended Data Fig. 4b,c). It is therefore possible that the higher baseline immunogenicity of these clones is due in part to greater surface presentation of neoantigens. Consistent with MMRd-derived neoantigens underlying the immunogenicity and ICB responsiveness of the clones, mice transplanted individually with single-cell clones derived from the low TMB control line (13-1) were uniformly and completely unresponsive to ICB (Extended Data Fig. 4f). Altogether, these results establish a model of MMRd wherein increasing mutational clonality (autochthonous tumors < cell lines < mixture of clones < clones) correlates with immunogenicity and ICB response.

Given that the parental cell line 09-2 arose in an immunocompetent host yet contained highly immunogenic subclones (M3, M7 and M8), we reasoned that tumors evade immunosurveillance not only by selective outgrowth of nonimmunogenic subclones but passively through the failure of immunity to eliminate otherwise immunogenic tumor cells present at a low clonal fraction. To test this hypothesis, we collected lung tumors and metastases from mice orthotopically transplanted with an equal mixture of clones M1–8 and reconstructed their clonal makeup by ultradeep, targeted amplicon sequencing of unique clone-defining SNVs (Methods, Fig. 4c and Extended Data Fig. 4g,h). Despite the immunogenicity of M3, M7 and M8, these clones were detected in 12 of 20 tumors analyzed from immunocompetent animals at clonal fractions ≥1%. Of these 12 animals 5 also developed liver metastases, 3 of which were clonal outgrowths of an immunogenic clone (M8) (Extended Data Fig. 4h). In contrast, none of the immunocompetent animals transplanted with M3, M7 or M8 alone formed metastases. This pattern was not substantially different to that of transplants in continuously T cell-depleted or ICB-treated mice. However, tumors from T cell-depleted mice on average showed significantly lower clonal diversity with greater dominance by individual clones (Fig. 4c,d). In addition, tumors from immunocompetent animals harboring one or more immunogenic clones (M3, M7 or M8) at ≥1% fraction were significantly more heterogeneous than tumors without these clones present (Fig. 4e). These results are in agreement with experiments in the autochthonous models (Fig. 3h–k and Extended Data Fig. 3h–n) and further support a revised understanding of immunosurveillance as a process that strongly selects against clonal, but not subclonal, neoantigens.

---

**Fig. 2 | MMRd models of lung and colon cancer are not immunogenic.**
**a,b**, Percentage lung area occupied by tumors of grades 1–4 (G1–4) in KP; $Msh2^{flox/flox}$ (KPM) and KP models at 5 (**a**) and 15 weeks (**b**) post-initiation with Cre-expressing adenovirus (SPC-Cre), with 16 and 15 animals 5 weeks post-initiation and 10 and 12 animals 15 weeks post-initiation. Normal lung and tumors were quantified using an automated CNN developed with Aiforia. **c,d**, Representative H&E and CNN annotations of tumor-bearing lungs from KPM (**c**) and KP (**d**) animals in **b**. Yellow is for normal lung, red for G1, green for G2, blue for G3 and orange for G4. Scale bar, 5 mm. **e–g**, IHC staining and Aiforia CNN quantification of T cell subsets in KPM and KP tumor-bearing lungs of animals in **a** and **b**. Representative IHC staining of lung tumor from an animal in **b** (left) with Aiforia CNN annotations (right) for CD4+ (green), CD8+ (yellow) and CD4+FOXP3+ T_reg cells (purple) (**e**). Scale bar, 100 μm. CNN quantification of IHC staining within lung tumors from KPM and KP animals is shown at 5 (**f**) and 15 weeks (**g**) post-initiation. **h,i**, Preclinical trial design in KPM and KP models (**h**) and

treatment arms (**i**). **j**, Change in solid lung volume as measured by μCT pre-treatment (10 weeks) and post-treatment (14 weeks). **k**, Lung tumor burden at necropsy (14 weeks) as measured by manual annotation of H&E-stained whole lung sections. **l**, Brightfield and fluorescent colonoscopy images of sgMsh2-targeted colon tumor, representing 16 animals. **m**, Change in colon tumor size by colonoscopy pre-treatment (20 weeks) and post-treatment (24 weeks) ($n = 23$ sgMsh2- and 7 sgCtl-targeted animals treated with ICB). **n**, Colon tumor burden at necropsy (24 weeks) as measured by stereomicroscopy ($n = 23$ sgMsh2- and 7 sgCtl-targeted animals treated with ICB (αPD-1/CTLA-4) and 10 sgMsh2-targeted animals treated continuously with T cell-depleting antibodies (αCD4/8)). Boxplots display median and interquartile range (IQR; box bounds), with whiskers extending to most extreme points (≤1.5× IQR) and all datapoints. Significance in **a**, **b**, **f**, **g**, **k** and **n** was assessed using Wilcoxon's rank-sum test with correction for multiple comparisons in **a**, **b**, **f**, **g** and **k**. P values in **n** are uncorrected.

## Neoantigen clonality tunes the T cell response

To gain further mechanistic understanding of how T cells shape clonal architecture, we first profiled the MHC-I immunopeptidome[45] of M1–8 clones (Supplementary Table 2). Using quantitative tandem mass spectrometry (MS–MS), we identified seven neoepitopes with

relative abundance patterns consistent with the associated mutations in the clones. Of these, five neoepitopes (71%) exhibited immunogenicity in normal mice after dendritic cell prime/boost/boost vaccination, by IFN-γ enzyme-linked immune adsorbent spot (ELISpot) and/or MHC-I:epitope tetramer staining of splenocytes (Extended Data

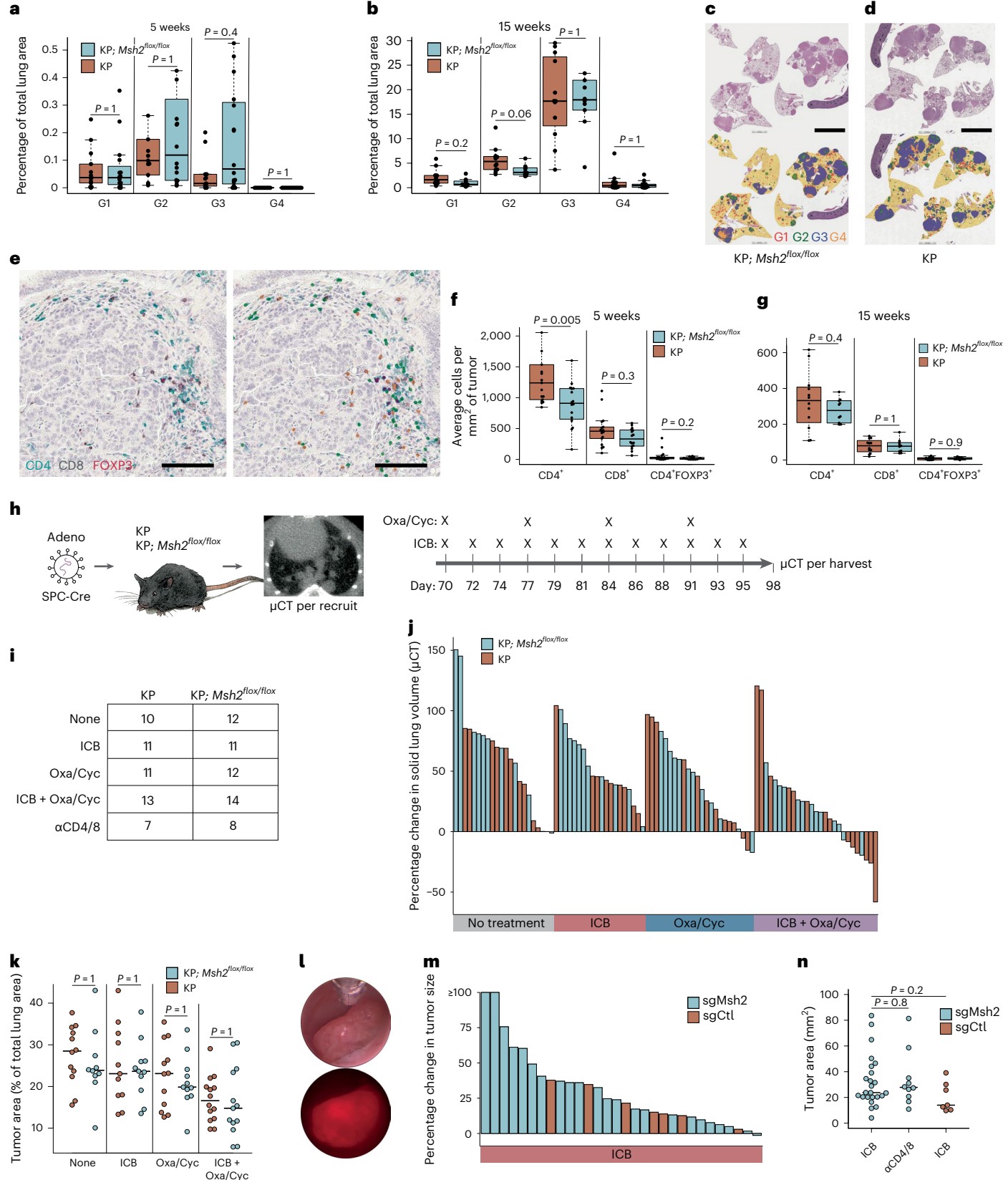

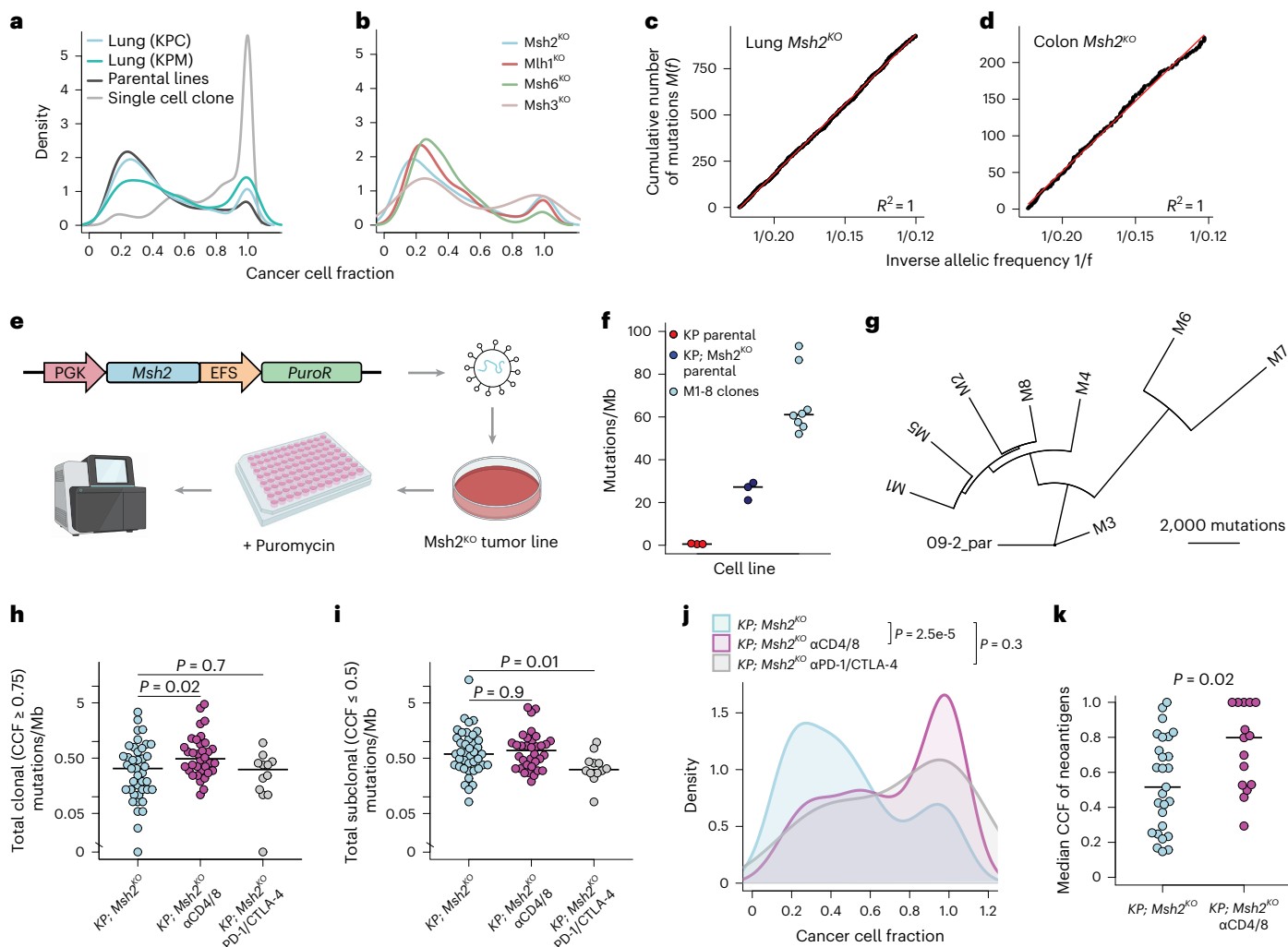

**Fig. 3 | MMRd models are defined by extensive ITH. a,b,** Distribution of CCF estimates of all SNVs in lung tumors, cell lines and clones M1–8 (**a**) and colon tumors (**b**). Smoothing was performed by Gaussian kernel density estimation. **c,d,** Cumulative distribution function of subclonal mutation evolution $M(f)$, as described in Williams et al.[44], for $Msh2^{KO}$ lung (**c**) and colon (**d**) tumors. $M(f) = \mu/\beta(1/f - 1/f_{max})$, where $f$ is the VAF/purity, $\mu$ the rate of somatic mutations and $\beta$ the fraction of cell divisions where both lineages survive. Linear distribution of $1/f$ (red line) is consistent with a neutral model of evolution. **e,** Schematic of single-cell cloning workflow with re-expression of $Msh2$. **f,** Total mutations identified in ex vivo lung tumor-derived cell lines and clones, as mutations per Mb of DNA ($n = 3$ sgCtl, 3 sgMsh2 and 8 clonal lines (M1–8)). Par, parental cell line. **g,** Phylogenetic tree of clonal interrelationships of M1–8 clones,

rooted on the parental line 09-2 and constructed using shared mutations with the parsimonious ratchet method. **h,i,** Total clonal (CCF ≥ 0.75) (**h**) and subclonal (CCF ≤ 0.5) (**i**) mutations per Mb in 16- to 20-week $Msh2^{KO}$ autochthonous lung tumors ($Msh2^{flox}$- and sgMsh2-targeted models) from animals with no treatment (light blue, $n = 41$), continuous antibody-mediated T cell depletion (αCD4/8, magenta, $n = 34$) and 4 weeks of ICB (αPD-1/αCTLA-4, gray, $n = 12$). **j,k,** CCF distribution (**j**) and per tumor median (**k**) of all expressed SNV-derived neoantigens in lung tumors from **h** and **i**. The significance was assessed using the two-sided Kolmogorov–Smirnov test. Smoothing in **a**, **b** and **j** was performed by Gaussian kernel density estimation. The significance in **h**, **i** and **k** was assessed using Wilcoxon's rank-sum test. $P$ values in **h–j** are uncorrected.

---

Fig. 5a–e). One neoepitope unique to the M5 clone, QAYAFLQHL, elicited a higher-magnitude, antigen-specific CD8⁺ T cell response than the highly immunogenic neoantigen SIINFEKL, from chicken ovalbumin[46] (Extended Data Fig. 5d). Although MS immunopeptidomics is far from exhaustive[47], our identification of five bona fide immunogenic neoepitopes supports the notion that neoantigens underlie the enhanced immunogenicity of $Msh2^{KO}$ clones (Fig. 4b and Extended Data Fig. 4f).

Next, we transplanted the M5 clone into the lungs of syngeneic mice at different clonal fractions, diluted with the other M1–8 clones, and analyzed the M5-specific CD8⁺ T cell response using flow cytometry with MHC-I:QAYAFLQHL dextramers (Fig. 5a). Consistent with sensitivity of the M5 clone to ICB (Fig. 4b), a robust QAYAFLQHL-specific T cell response was induced in lungs and mediastinal draining lymph nodes

(mLNs) of clonally transplanted mice after 2 weeks of ICB treatment. As the clonal fraction and total number of M5 cells were decreased, however, the magnitude of this response also decreased (Fig. 5b–d and Extended Data Fig. 5f–h). Similar results were obtained in analogous experiments with the M2 clone using MHC-I:AALQNAVTF tetramers to analyze M2-specific CD8⁺ T cells (Extended Data Fig. 5i–k). Surprisingly, the quality of the QAYAFLQHL-specific T cell response also decreased with lower clonal fraction, with a significantly smaller percentage of QAYAFLQHL-specific T cells expressing the major effector protease of cytotoxic T cells, granzyme B (GZMB) and a significantly greater percentage expressing TCF1 (Fig. 5e–i and Extended Data Fig. 5l–p). Expressed highly in naive and early activated T cells, the transcription factor TCF1 is lost during effector differentiation[48]. We have previously shown that TCF1⁺GZMB⁻ tumor-specific T cells are enriched

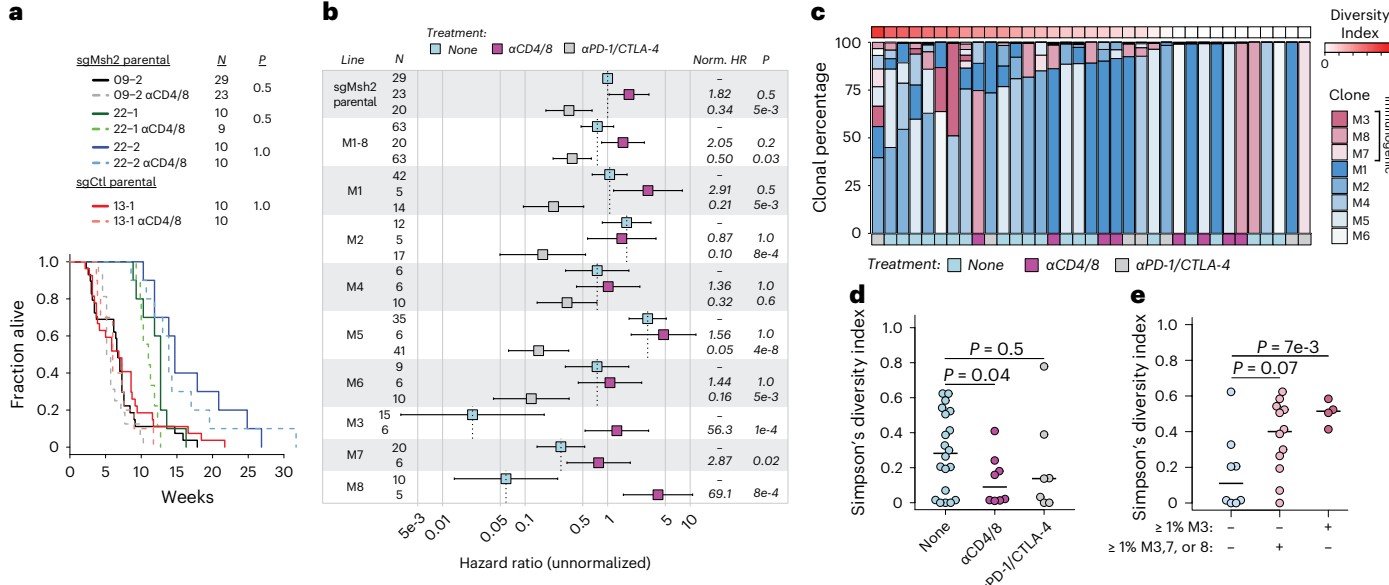

**Fig. 4 | ITH enables immune evasion of MMRd tumors. a,b,** Survival of syngeneic mice orthotopically transplanted via intratracheal instillation with indicated lung tumor cell lines and clones. **a,** Kaplan–Meier curves of mice transplanted with parental sgMsh2- and sgCtl-targeted parental lines, with and without continuous T cell depletion (αCD4/8: lighter shades and dotted lines). **b,** HRs for mice transplanted with parental sgMsh2 line (09-2), an equal mixture of M1–8 clones and individual M1–8 clones, with and without continuous αCD4/8 and ICB treatment. Norm. HR is normalized HR, which was calculated by dividing plotted HRs of each treatment group by the HR of the no treatment group (reference) for each line separately. Bars represent upper and lower 95% confidence intervals. **c,** Estimation of clonal percentages of M1–8 clones in lung tumors from animals transplanted with an equal mixture of all clones and receiving no treatment (n = 20), continuous αCD4/8 (n = 8) or ICB (n = 7).

Clonal percentages were determined by targeted deep amplicon sequencing of four private SNVs per clone. Diversity is Simpson's diversity index based on proportions of M1–8 clones present in tumors. **d,e,** Simpson diversity index of tumors between treatment groups (**d**) and across tumors containing no immunogenic clones, ≥1% M3, M7 or M8 or ≥1% M3 (**e**). ICB treatment in **b–e** was started 2 weeks post-transplantation and continued for 4 weeks. Shades of blue and red in **c** denote baseline (no treatment) nonimmunogenic and immunogenic lines, respectively, with otherwise no significant differences indicated by color. Significance in **a** and **b** was assessed using Cox's proportional hazards regression with Holm's correction for multiple comparisons of two hypotheses: no treatment versus αCD4/8 = 13 tests and no treatment versus αPD-1/CTLA-4 = 12 tests. Significance in **d** and **e** was assessed using Wilcoxon's rank-sum test and uncorrected P values are shown.

and characteristic of tolerogenic dysfunction in a model of microsatellite stable colon cancer[49]. It is interesting that transplantation of an equal number of M5 cells (100,000) at clonality versus CCF of 0.5 (with 100,000 M1–4,6–8 cells) resulted in no significant difference in magnitude, GZMB+ or TCF+ percentage of the T cell response (Fig. 5d,g,i and Extended Data Fig. 5h,n,p), suggesting that total mass of neoantigen-expressing tumor cells is the primary determinant of quality of the neoantigen-specific T cell response. Altogether, these results provide a compelling rationale for why immunosurveillance fails to delete tumor cells bearing subclonal neoantigens (Fig. 5j).

## Clonal TNB predicts ICB response in human MMRd cancer

To explore the translational relevance of our findings, we reanalyzed sequencing data from two clinical trials of anti-PD-1 treatment in advanced MMRd colorectal cancer (CRC; Bortolomeazzi et al.[50]) and gastric cancer (Kwon et al.[35]), including 16 and 13 patients, respectively. Predicting neoantigens and the associated CCFs (Supplementary Table 3), we asked whether clonal TNB and ITH are associated with the response to anti-PD-1. Clonal (CCF ≥ 0.75), but not subclonal (CCF ≤ 0.5), TNB was significantly associated with objective response (OR), whereas high ITH index (subclonal to clonal neoantigen ratio) was significantly associated with nonresponse (NR) (Fig. 6a,b and Extended Data Fig. 6a–d). Total TNB was also significantly associated with OR (Extended Data Fig. 6e–g), probably because tumors in these studies had generally more clonal than subclonal neoantigens. Although clonal TNB was generally correlated with total TNB, two notable outliers, both nonresponders, had high total TNB but very low clonal TNB (Extended Data Fig. 6h).

Clonal, but not subclonal, TNB was also significantly associated with longer progression-free survival (PFS) in combined analysis of the trials, whereas the ITH index was associated with shorter PFS (Fig. 6c–e). Total TNB also correlated with longer PFS, but did not reach the same level of significance as clonal TNB (Extended Data Fig. 6i). Importantly, there was no significant difference in PFS between the two trials (Extended Data Fig. 6j), justifying their combination in these analyses. Overall, these results support the major conclusions from our mouse models and suggest that clonal TNB is a more accurate predictor of ICB response than overall TMB in MMRd cancers. Although the trials that we reanalyzed in the present study are limited by small sample size and will require validation in larger prospective investigations in MMRd cancers, which are currently lacking, our results align with a growing body of literature that supports the superior predictive power of clonal versus total TMB/TNB as a biomarker of ICB response across other cancers[20–26].

## Discussion

In this Article, we report sporadic MMRd mouse models that recapitulate critical features of human lung and colon cancer, including genetics, histopathology and in situ initiation in the relevant tissue microenvironment. Unlike previous studies that demonstrated a role for MMRd and TMB in immunotherapy response, in which mutagenesis occurred in vitro[26–28], our models enable study of mutations continuously acquired in vivo from tumor initiation through advanced disease. Other studies showed immunotherapy efficacy in autochthonous cancer models employing tissue-specific knockout of *Msh2* or activation of mutant *Pole* (*Pole^P286R*) during embryogenesis[51,52]. These previous models

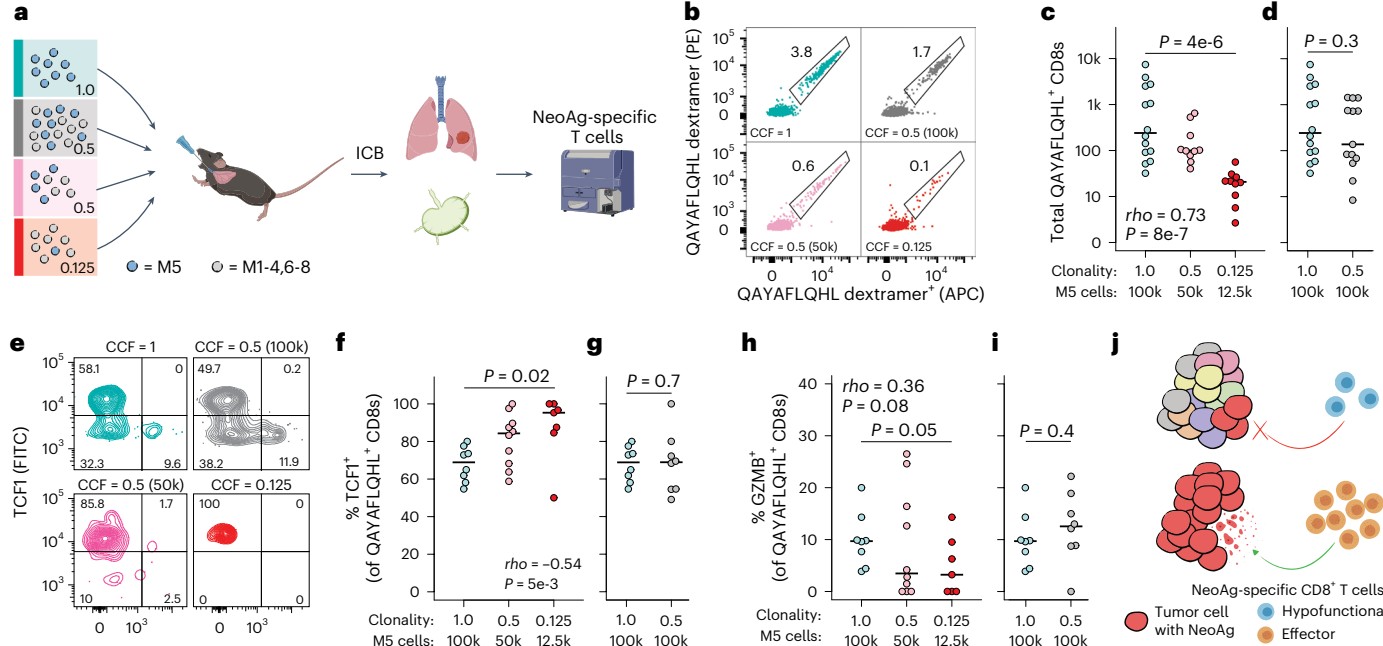

**Fig. 5 | The quality of tumor-specific T cell response is tuned by neoantigen clonality.** Flow cytometric analyses of M5 clone neoantigen (QAYAFLQHL)-specific T cells isolated from lungs and mLNs of syngeneic mice intratracheally transplanted with M5 at different CCFs, diluted using M1–4,6–8 clones. Mice were treated with ICB (αPD-1/CTLA-4) for 2 weeks starting 2 weeks post-transplantation. Light blue shows CCF = 1 (100,000 M5 cells), *n* = 14 animals; gray shows CCF = 0.5 (100,000 M5 + 100,000 M1–4,6–8 mixed cells), *n* = 13 animals; pink shows CCF = 0.5 (50,000 M5 + 50,000 M1–4,6–8 mixed cells), *n* = 10 animals; and red shows CCF = 0.125 (12,500 M5 + 87,500 M1–4,6-−8 mixed cells), *n* = 10 animals. **a**, Outline of experimental design. **b–d**, Representative flow plot (**b**) and total QAYAFLQHL-specific CD8+ T cells (**c** and **d**) in mLNs as determined by MHC-I dextramer staining in two channels (PE and APC). **e–i**, Representative flow plot (**e**) and quantification of percentage QAYAFLQHL-specific CD8+ T cells positive for TCF1+ (**f** and **g**) and GZMB+ (**h** and **i**) in mLNs. **j**, Proposed mechanism of failure of neoantigen (NeoAg)-specific T cells to delete subclonal targets. The colors of the tumor cells (left) represent distinct neoantigen profiles. Blue and orange T cells (right) represent poor versus productive effector differentiation, respectively. Significance in **c**, **f** and **h** was assessed using both Spearman's rank correlation with a numeric *x* axis (CCF) and Wilcoxon's rank-sum test (CCF = 1 versus 0.125 groups). The significance in **d**, **g** and **i** was assessed using Wilcoxon's rank-sum test. Samples with <10 QAYAFLQHL-specific CD8+ T cells detected during flow cytometric acquisition were excluded from analysis in **f–i**.

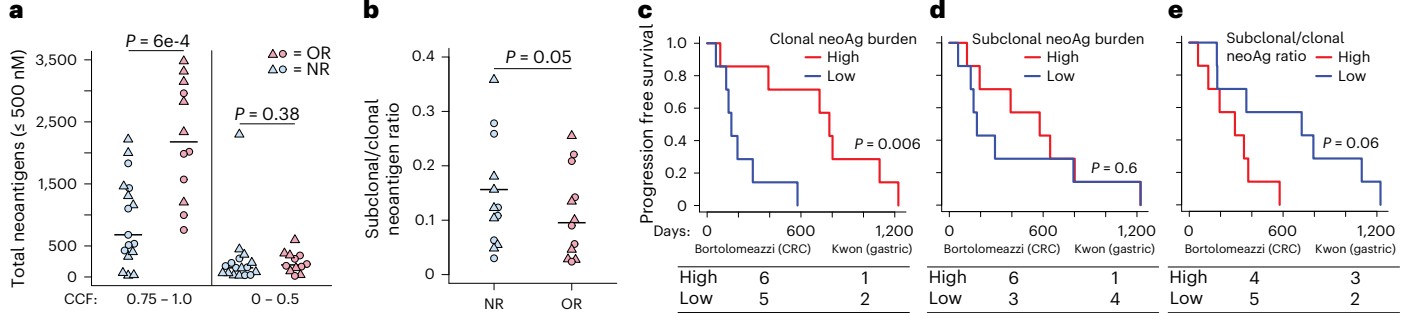

**Fig. 6 | Clonal neoantigen burden is predictive of ICB response in human MMRd cancer.** Meta-analysis of neoantigen burden, clonality and response to anti-PD-1 treatment in clinical trials of MMRd gastric (Kwon trial[35], *n* = 13 patients) and colorectal (Bortolomeazzi trial[50], *n* = 16 patients) cancer. **a,b**, Total clonal (CCF = 0.75–1.0) and subclonal (CCF = 0–0.5) neoantigen burden (**a**) and subclonal to clonal neoantigen ratio (**b**) in patients with objective response (OR, partial or complete) versus nonresponse (NR). **c–e**, PFS of patients in the upper versus lower quartiles of clonal neoantigen burden (**c**), subclonal neoantigen burden (**d**) and subclonal to clonal neoantigen ratio (**e**). Number of patients from each study in upper (high) and lower (low) quartiles is indicated under the plots. The significance in **a** and **b** was assessed using Wilcoxon's rank-sum test and in **c–e** using Cox's proportional hazards regression with the clinical trial study as a covariate.

recapitulate familial cancers like Lynch syndrome in the accumulation of mutations in normal parenchyma preceding transformation (<5% of colorectal cancers (CRCs)), but not sporadic loss of MMR in established tumors (10–15% of CRCs)[14,15]. In contrast, our models, which more closely resemble sporadic MMRd, followed a model of neutral evolutionary dynamics[44] and did not display increased baseline immunogenicity or response to ICB, probably owing to timing of MMR inactivation and the resulting patterns of clonal expansion[53]. We induced MMRd concomitantly with tumor initiation, resulting in mutation accumulation during

exponential cellular expansion that is reminiscent of so-called 'born to be bad' colon tumors that follow an early explosive growth trajectory[54]. It is interesting that it was recently shown that MMRd occurring either de novo or induced by temozolomide treatment in advanced glioma led to extensive ITH and poor ICB response[38].

Tumors in our models developed extensive ITH and a high burden of subclonal mutations that was not detected by bulk sequencing analysis, highlighting the importance of standardization of clinical pipelines to estimate TMB. Strategies to robustly assay ITH, such as multi-region

or single-cell DNA-sequencing, may enhance the predictive utility of TMB. Overall, the results from our models strongly support a potential role of ITH in the failure of ICB in some patients with MMRd cancer. However, studies in the mouse cannot be generalized to humans and these results will require clinical validation. Our reanalysis of MMRd cancer clinical trials[35,50] showed significant association of clonal neoantigen burden and ITH index with ICB response and lends credibility to our models, but is limited by small sample size and not powered to differentiate any predictive value of subclonal neoantigen burden or ITH index beyond their association with clonal neoantigen burden. Larger prospective clinical studies will be required to definitively establish the role of ITH and its potential utility as a biomarker of ICB response in human MMRd cancers[5,6].

Similar to other mechanistic studies of ITH[19,26], we found that experimental reconstitution of ITH potentiated immune evasion. Recently, a preclinical study showed that genetic or pharmacological enrichment of MMRd in the context of mixed MMRd/MMR-proficient (MMRp) cell-line transplants potentiated rejection of the MMRp fraction[55]. This must be interpreted with care, however, because both MMRd and MMRp fractions were derived from the same carcinogen (*N*-nitroso-*N*-methylurethane)-induced colon carcinoma line, CT26, and probably share a high burden of clonal neoantigens that may underlie rejection of the MMRp fraction. What distinguishes our study from these prior cell line-based studies is that mutagenesis occurred spontaneously entirely in vivo. That some of the subclones we isolated were highly immunogenic on re-transplantation at clonal, but not subclonal, fraction suggests that they were protected from deletion by high ITH in the original tumor. This probably occurs passively due to low cellularity precluding efficient crosspriming and driving early T cell dysfunction or ignorance[19,49]. Indeed, our high-resolution analysis of neoantigen-specific T cells after ICB treatment showed that the magnitude and effector potential of the response are attenuated with decreasing neoantigen clonality. This is in agreement with an orthogonal study that used retroviral neoantigen libraries to manipulate clonal fraction[19]. However, although that study concluded that neoantigen clonal fraction, not total cellularity, is the major determinant of the response, our experiments suggest the converse. Additional studies are needed to resolve these and other complexities, including the interplay of ITH with T cell interclonal dynamics, where the immune response may be deployed against a limited subset of 'dominant' neoantigens[56], at the expense of recognition of lower affinity, poorly expressed or subclonal neoantigens.

Paradoxically, we found that immunosurveillance may exacerbate ITH by shaping the clonal architecture of tumors while failing to delete most neoantigens. Therefore, we conclude that ITH is shaped by the interplay of tumor intrinsic, positively selective and immunogenic, negatively selective evolutionary pressures. These results provide nuance to our understanding of immunoediting and highlight the power of our models to capture a hallmark of human cancer that is lacking in carcinogen-induced and genetically engineered models alike[57]—the gradual accumulation of mutations over time[36].

Our results raise important questions related to therapies aimed at deliberately increasing TMB to enhance tumor immunogenicity[27,58]. These strategies will probably fail to elicit meaningful immune engagement. More concerning, collateral mutagenesis may drive more aggressive cancer, therapy resistance or secondary malignancies. Future studies with models that enable temporal control of cooperating tumorigenic events will be helpful in determining the impact of cancer clonal selection on immunosurveillance and immunotherapy response.

## Online content

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

## Methods

### Mice

All animal use was approved by the Department of Comparative Medicine at the Massachusetts Institute of Technology (MIT) and the Institutional Animal Care and Use Committee under protocol no. 0714-076-17. Mice were housed with a 12-h light/12-h dark cycle with temperatures in the range 20–22 °C and 30–70% humidity. $Kras^{LSL-G12D}$ (ref. [59]); $Trp53^{flox/flox}$ (ref. [60]); $R26^{LSL-Cas9}$ (ref. [61]) (KP;$R26^{LSL-Cas9}$) mice were maintained on an F1 (C57BL/6 × 129/SvJ) background. $Kras^{LSL-G12D}$; $Trp53^{flox/flox}$; $Msh2^{flox/flox}$ (ref. [32]) (JAX, stock no. 016231) and $R26^{Cas9}$ (ref. [62]) (JAX, stock no. 028555) mice were maintained on a pure C57BL/6 background. Lung cell lines were isolated from tumors induced in albino C57BL/6 hosts chimeric for tissue derived from blastocyst injection of a $KP;R26^{LSL-Cas9}$ embryonic stem (ES) cell line (12A2) of mixed C57BL/6 and 129/SvJ background and male sex, as previously described[61]. In orthotopic lung studies, cell lines were transplanted into male chimeras generated from the same 12A2 ES cell line at 10–16 weeks of age. These chimeras are tolerized to C57BL/6 and 129/SvJ tissues, potential antigens in the $R26^{LSL-Cas9}$ allele and PuroR introduced into cell lines with $Msh2$ re-expression (unrecombined $Kras^{LSL-G12D}$ expresses PuroR). Autochthonous tumors in lung and colon were induced in approximately equal numbers of male and female mice at 8–16 weeks of age.

### Tumor models

Tumor burden, where measurable, was not allowed to exceed 1 cm$^2$ and animals showing discomfort or distress were humanely euthanized following the recommendations of the American Veterinary Medical Association. Autochthonous lung tumors in $Kras^{LSL-G12D}$; $Trp53^{flox/flox}$; $R26^{LSL-Cas9}$ and $Kras^{LSL-G12D}$; $Trp53^{flox/flox}$; $Msh2^{flox/flox}$ mice were induced by intratracheal instillation of $2 \times 10^4$ transduction units (TU) of lentivirus and $2 \times 10^8$ plaque-forming units of adenovirus-expressing Cre driven by the alveolar type II cell-specific surfactant protein C promoter (SPC-Cre), respectively, as previously described[31]. Autochthonous colon tumors in $R26^{Cas9}$ mice were induced by endoscope-guided submucosal injection in the distal colon, as previously described[33,63]. Two injections at $1.5 \times 10^6$ TU of lentivirus in 50 µl of Opti-MEM were delivered per mouse. Lentivirus was produced in HEK293 cells (American Type Culture Collection) and concentrated as previously described[31], and functional titers (Cre activity, $mScarlet$ fluorescence) measured as previously described[64]. Cell lines were orthotopically transplanted by intratracheal instillation of $1 \times 10^5$ cells in 50 µl of Spinner Modification of Minimal essential Eagle's Medium (SMEM)/5 mM EDTA, followed by a 30-µl rinse with the same medium. Cell lines were established from autochthonous lung tumors by microdissection and mechanical mincing in digestion buffer (Hanks' balanced salt solution with 1 M Hepes, 125 units ml$^{-1}$ of collagenase type IV (Worthington) and 20 µg ml$^{-1}$ of DNase (Sigma-Aldrich)), followed by incubation at 37 °C with gentle agitation for 30 min and plating in RPMI + 10% fetal bovine serum (FBS). Lines were plated into 50:50 RPMI/Dulbecco's modified Eagle's medium (DMEM) + 10% FBS at first passage and DMEM + 10% FBS at second passage and thereafter. Cells were taken for WES at the third passage. $Msh2$-expressing lentivirus was produced as above. Cells were incubated with lentiviral supernatant and 3 d later selected and maintained thereafter on medium with 6 µg ml$^{-1}$ of puromycin (Thermo Fisher Scientific).

### WES and mutation calling

Whole-exome libraries were generated using the SureSelect XT Mouse All Exon (Agilent) target enrichment kit; 100-bp paired-end sequencing of samples was performed on the Illumina HiSeq 4000 platform, with the exception of M1–8 passage of 20 single-cell clones, which were 150-bp paired-end sequenced on the Illumina NovaSeq 6000 S4 platform. Library preparation and sequencing to 100× on-target coverage were performed by Psomagen. Raw sequencing reads were mapped to the GRCm38 build of the mouse reference genome using BWA-MEM v.0.7.17-r1188 (ref. [65]). Aligned reads in BAM format were processed following the Genome Analysis Toolkit (GATK) v.4.1.8.0 Best Practices workflow to remove duplicates and recalibrate base quality scores[66]. Median coverage was 98 (25th quartile = 86; 75th quartile = 112), 90 (25th quartile = 82; 75th quartile = 108) and 103 (25th quartile = 100; 75th quartile = 107) for normal tails, autochthonous tumors and cell lines, respectively.

Somatic SNVs and indels were detected using Mutect2, MuSE v.1.0rc[67], VarDict v.1.8.2 (ref. [68]) and Strelka2 v.2.9.2 (ref. [69]) against matched normal tails. Mutect2 was run using a panel of normals compiled from the 18 tails analyzed in the present study. Each caller was run independently on each tumor-normal pair and calls were integrated using SomaticCombiner v.1.03 (ref. [70]). For colon tumors, a panel of four normal tails was used to generate the matched normal control for all samples, because these mice were of pure background. We considered SNV mapping to only exonic regions that were detected by Mutect2 and at least one of the other algorithms. To increase accuracy of indel detection, only indels detected by at least two algorithms were considered. Variants mapping to dbSNP (build ID 150) positions were discarded. Mutations identified in tumors from two or more animals or in at least 50% of tumors from the same animal were discarded. No VAF filter was applied. Microsatellites were annotated using SciRoKo v.3.4 (ref. [71]), with minimum score = 8, seed length = 8, repeats = 2 and mismatch penalty = 1.

### CCF estimation

Somatic copy-number aberrations were detected by integrating output of GATK and FreeBayes v.1.3 (ref. [72]) using PureCN v.1.16.0 (ref. [73]). Briefly, GATK4 Somatic CNV workflow was utilized for normalization of read counts and genome segmentation using the panel of normals from all tails. FreeBayes was used to obtain B-allele frequencies for dbSNP variant sites. PureCN was used to integrate output from GATK and FreeBayes to estimate allele-specific consensus copy-number profile, purity and ploidy of each sample. The ploidy of cell lines was determined experimentally by metaphase spreads and input into PureCN. Finally, the CCF for each SNV and indel was computed using the R (v.4.0.2) package cDriver[43].

### Mutational signature and MSI analyses

Mutational signatures were extracted with the R (v.4.0.2) package MutationalPatterns[74] (v.3.2.0) using the COSMIC Mutational Signatures catalog v.3 (ref. [75]). We used the function fit_to_signatures with default parameters and included only those mutational processes known to operate in human colon and/or lung cancer (excluding tobacco smoking)[75]: SBS1, SBS5, SBS6, SBS10a, SBS10b, SBS14, SBS15, SBS17a, SBS17b, SBS18, SBS21, SBS26, SBS28, SBS37, SBS40 and SBS44. For visualization, we collapsed signatures of MMRd (SBS6, SBS14, SBS15, SBS21, SBS26 and SBS44; labeled as MMRd) and POLE deficiency (SBS10a, SBS10b, SBS28 and SBS17b; labeled as POLE). Goodness of fit was determined by computing cosine similarity between observed and reconstructed mutational spectra using estimated signature contributions. To estimate the contribution of each mutational process to the human MSI CRC mutational spectrum, we analyzed somatic mutations in the Kwon et al. cohort (non-formalin-fixed paraffin-embedded samples)[35] following the methodology described above. Sample MSI score was calculated using MSIsensor-pro (v.1.2.0)[34] against matched normal tails.

### Clonal deconvolution by targeted amplicon sequencing

To identify private somatic SNVs for distinguishing individual clones in the M1–8 mixed clone tumors (Fig. 4c and Extended Data Fig. 4g,h), we compiled all clonal SNVs in copy-number-neutral regions (four copies, as all lines were tetraploid by metaphase spreads). We then checked the BAM files across all other samples for the complete absence of reads supporting the alternative allele (base quality >20, mapping quality >30) using an in-house Python script relying on the Pysam library. Four

private SNVs for each clone and four common SNVs were validated by PCR amplification and Sanger sequencing before proceeding. The 200- to 250-bp regions spanning these SNVs were either individually PCR amplified from samples, gel purified and combined, or amplified in parallel using a multiplexed PCR panel with primers carrying unique molecular indices (CleanPlex UMI Custom Panel, Paragon Genomics). Amplicon libraries were sequenced on the Illumina NovaSeq 6000 S4 platform with 150-bp paired-end chemistry.

Reads were aligned to a fasta reference file of all targets (±250 nt upstream/downstream of SNV, GRCm38) using BWA-MEM (v.0.7.17-r1188)[65], following the GATK Best Practices workflow. Pile-ups were generated using the mpileup function of bcftools v.1.10.2 (ref. [76]) with --min-BQ 30 and a bed file of all SNV coordinates. For Clean-Plex UMI libraries, the following functions in fgbio (v.2.0.1) (https://github.com/fulcrumgenomics/fgbio) were called to extract unique molecular identifiers (UMIs) and call consensus reads: ExtractUmis-FromBam, GroupReadsByUmi and CallMolecularConsensusReads. Using a customized R (v.4.0.2) script, total and SNV-specific depths at all locations were extracted. All SNVs were supported by more reads than other alternative alleles in the M1–8 clone-equal mixture control, except for M6_2, which was excluded from subsequent analysis. Background PCR/sequencing error for each SNV was estimated using the median observed frequencies of SNVs in all metastases of different clones, which represented truly clonal controls. SNV frequencies were adjusted by subtracting background values. Clonal percentages in ex vivo tumors were estimated by taking the median of private SNV frequencies, multiplying by 4 (SNVs are $1/4n$) and dividing by tumor purity—estimated as the median observed/expected ratio of frequencies of the four common SNVs (present in all clones).

### Neoantigen prediction and expression

Variant consequence was annotated using Ensembl Variant Effect Predictor (VEP) v.99 (ref. [77]) with Wildtype and Downstream plugins, the VEP cache and reference genome for GRCm38, and the following parameters: --symbol, --terms=SO, --cache, --offline, --transcript_version and --pick. The --pick parameter was reordered from default to report the transcript with most extreme consequence for each variant: rank, canonical, appris, tsl, biotype, ccds, length and mane. Neoepitopes were predicted with C57BL/6 mouse MHC-I alleles, *H2-K1* (H-2K$^b$) and *H2-D1* (H-2D$^b$) and variant effect predictions using pVACtools v.1.5.7 (ref. [78]). Mutant peptides were generated for peptides that were 8–11 amino acids and MHC:peptide binding affinity was predicted for all peptide:MHC allele pairs with NetMHC-4.0, NetMHCpan-4.0, SMM v.1.0 and SMMPMBEC v.1.0 (refs. [79–82]). The median value across all affinity predictions was taken as the final measure of binding affinity. Neoantigens were subset to those with median predicted H-2K$^b$/H-2D$^b$ affinity ≤500 nM. Where multiple neoantigens were predicted for the same SNV, only that with highest predicted affinity was retained.

To assess allele-specific expression of neoantigens, RNA-sequencing (RNA-seq) was performed on autochthonous lung tumors (10 sgMsh2 and 10 sgMsh2 with αCD4/8 treatment) and M1–8 clones. Complementary DNA libraries were prepared using Kapa mRNA Hyperprep and 150-bp paired-end sequencing was performed on the Illumina NextSeq platform. Reads were aligned to the reference genome (GRCm38) using STAR v.2.7.1a[83] with outFilterMultimapN-max = 20, alignSJoverhangMin = 8, alignSJDBoverhangMin = 1, outFilterMismatchNmax = 999, outFilterMismatchNoverLmax = 0.1, alignIntronMin = 20, alignIntronMax = 1,000,000, alignMatesGap-Max = 1,000,000, outFilterScoreMinOverLread = 0.33, outFilter-MatchNminOverLread = 0.33 and limitSjdbInsertNsj = 1,200,000. PCR duplicates were removed using Picard v.2.23.4 (ref. [84]). Considering somatic variants identified by WES, we used a customized Python script to interrogate the presence of these variants in the RNA-seq BAM files. Only nonduplicate reads with mapping quality ≥255 and bases with base quality ≥20 were considered to compute VAFs.

### Histology and immunohistochemistry

Quantification of lung tumor burden by grade was performed on scans of hematoxylin and eosin (H&E)-stained sections by automated convolutional neural network (CNN)—developed in collaboration with Aiforia Technologies Oy in consultation with veterinarian pathologist R. Bronson. Using semantic multi-class segmentation, the CNN was trained to classify lung parenchyma and adenocarcinoma grades 1–4. For supervised training, selected areas from 93 slides were chosen. The algorithm performed consistently and with high correlation with human graders across multiple validation datasets independent of the training dataset. Algorithm v.NSCLC_v25 was used. Triple staining (CD8a, CD4 and FOXP3) immunohistochemistry (IHC) and CNN quantification (Aiforia) were performed as previously described[49]. CD3 infiltration in single-stain slide scans was measured as percentage of pixels positive for stain (diaminobenzidine) in Aperio ImageScope. The area of positive and negative MSH2 staining was quantified by manual annotation in QuPath v.0.1.2 (ref. [85]).

### Western blotting

Cells were lysed in radioimmunoprecipitation (RIPA) buffer (Thermo Fisher Scientific), protein concentration determined using BCA Protein Assay (Thermo Fisher Scientific) and equal protein quantities (20–40 µg) run on NuPage 4–12% Bis–Tris gradient gels (Thermo Fisher Scientific) by sodium dodecylsulfate–polyacrylamide gel electrophoresis and transferred to poly(vinylidene fluoride) membranes. Western blotting was performed against MSH2 (catalog no. D24B5, Cell Signaling Technology) at 1:1,000, MLH1 (catalog no. ab92312, Abcam) at 1:1,000, glyceraldehyde 3-phosphate dehydrogenase (catalog no. 6C5, Santa Cruz) at 1:5,000 and β-actin (catalog no. 13E5, Cell Signaling Technology) at 1:5,000. Blots were stained with horseradish peroxidase (HRP) anti-rabbit immunoglobulin G and developed with Western Lightning Plus-ECL (Perkin Elmer) on X-ray film.

### In vivo antibody and chemotherapy dosing

Antibodies were delivered intraperitoneally in 100 µl of phosphate-buffered saline (PBS). αCD4 (catalog no. GK1.5, BioXCell) and αCD8 (catalog no. 2.43, BioXCell) were administered at 200 µg every 4 d. αPD-1 (catalog no. 29F.1A12, BioXCell) was administered at 200 µg 3× a week. αCTLA (catalog no. 9H10, BioXCell) was administered at an initial dose of 200 µg, with subsequent doses at 100 µg, 3× a week. Oxaliplatin (Sigma-Aldrich) and cyclophosphamide (Sigma-Aldrich) (Oxa/Cyc) were co-delivered intraperitoneally in 100 µl of PBS at 2.5 mg per kg body weight and 50 mg per kg body weight, respectively, once a week for 3 weeks, as previously described[42].

### In vivo tumor imaging and quantification

Lung tumor progression was monitored longitudinally by X-ray microcomputed tomography (µCT) using a GE eXplore CT 120 system, as previously described[86]. Solid lung volume (tumor burden) was quantified using a customized MATLAB (MathWorks) script, as previously described[86]. Colon tumor progression was monitored longitudinally using a Karl Storz colonoscopy system with white light and red fluorescent protein fluorescence and biopsy forceps serving as a landmark for objective positioning, as previously described[49].

### Lentiviral constructs

The U6::sgRNA-EFS::Cre (pUSEC) lentiviral construct[86] was digested with BsmBI and sgRNAs cloned as previously described[87]. H1::sgApc-U6::sgRNA-EFS::mScarlet was generated by Gibson assembly using an H1::sgApc-scaffold gBlock synthetic gene fragment (IDT), PCR amplicons of U6::BsmBI-filler-BsmBI-scaffold, elongation factor-1 (EFS) promoter and *mScarlet*[88], and a lentiviral backbone from the Trono laboratory (Addgene). This was then digested with BsmBI and a second sgRNA cloned as above. The sgRNA sequences, including previously published sgApc[33] and sgCtl (*mScarlet* targeting)[64] are provided

in Supplementary Table 4. The sgRNA controls against *Olfr102* and *mScarlet* were used interchangeably with no observed differences in tumorigenesis. PGK::Msh2-EFS::PuroR was generated by Gibson assembly using multiple gBlocks spanning murine *Msh2* (C57BL/6), PCR amplicons of PGK (3-phosphoglycerate kinase) promoter, EF-1 Alpha Short (EFS) promoter and *PuroR*, and the Trono lentiviral backbone. All primers were ordered from Sigma-Aldrich.

## Validation of CRISPR–Cas9 editing and estimation of tumor purity

To validate efficiency of gene editing by clustered regularly interspaced short palindromic repeats (CRISPR)–Cas9, 200- to 250-bp regions spanning sgRNA sites in the genome were amplified and deep sequenced (Massachusetts General Hospital DNA Core). Colon tumor purity was estimated using a non-wild-type allele fraction at the sgRNA-targeted site in *Apc*. Loss of *Apc* is prerequisite for tumorigenesis in the model and thus an assumption was made that all tumor cells harbor loss-of-function edits at this locus. Tumor purity in sgMsh2-targeted lung tumors was estimated using WES BAM coverage spanning exons of the *Trp53^flox* allele and flanking genes (*Wrap53*, *Atp1b2*), which were retrieved using the bedcov function of SAMtools v.1.10. The ratio of median coverage in flanking exons (*Wrap53* exons 0–9, *Trp53* exon 11 and *Atp1b2* exons 0–6) versus *Trp53* exons flanked by Cre loxP sites (exons 2–10) was calculated in tumors and normal tails. This ratio in tumors, representing the extent of *Trp53^flox* recombination, was then normalized to the median ratio across matched normal tails to estimate tumor purity, with the assumption that all tumor, but not normal, cells underwent complete recombination of *Trp53^flox* alleles. Efficiency of *Msh2* knockout in KP;*Msh2^flox/flox* lung tumors was similarly estimated by taking the ratio of reads at the exon flanked by loxP sites (exon 12) and surrounding exons, and adjusting this by tumor purity.

## In vitro cell-line assays

Serial live cell imaging of cell lines grown in 96-well plates (Corning) and quantification of confluence were performed with an IncuCyte S3 (Sartorius). Eight replicate wells were seeded with 100 cells and imaged every 3 h for ~6 d. Murine IFN-γ (PeproTech) was used for in vitro stimulation of cell lines for 24 h, followed by live/dead staining (ghost ef780 (Corning), 1:500) in PBS and surface staining in 1 mM EDTA, 25 mM Hepes, 0.5% heat-inactivated FBS in PBS with anti-H-2K^b allophycocyanin (APC) (catalog no. AF6-88.5.5.3, Thermo Fisher Scientific, 1:200), anti-H-2D^b FITC (catalog no. 28-14-8, Thermo Fisher Scientific, 1:200) and anti-PD-L1 phycoerythrin (PE)-Cy7 (catalog no. 10F.9G2, BioLegend, 1:200). Samples were run on a BD LSRFortessa using BD FACSDiva v.8.0 software. Results were analyzed in FlowJo v.10.4.2, excluding dead (ef780-positive) cells.

## Phylogenetic tree analysis

All somatic SNVs and indels called by the WES analysis pipeline in M1–8 clones and the 09-2 parental cell line were considered in constructing a phylogenetic tree. The R (v.4.0.2) Bioconductor package phangorn (v.2.7.0) was used to construct a tree from a binary presence/absence matrix of mutations across clones and 09-2_par. Specifically, the function prachet was used to calculate the tree using the parsimonious ratchet method and the function acctran was used to calculate branch lengths.

## MHC-I immunopeptidomics

MHC-I (H-2K^b and H-2D^b) peptide isolation was performed on $10^8$ cells per triplicate for each M1–8 clone as we have previously described[49]. Cells were grown to confluence before stimulation with 10 ng ml^−1 of murine IFN-γ (PeproTech) for 18 h before collection. Pulldowns were performed with 40 μl (bed volume) of rProtein A Sepharose beads (GE Healthcare) preloaded with 1 mg of anti-H-2K^b antibody (Y3, BioXCell) and 1 mg of anti-H-2D^b antibody (catalog no. 28-14-8S, purified from

HB-27 hybridoma), performed sequentially. Peptides were eluted in 500 μl of 10% acetic acid and purified with 10-kDa MWCO spin filters (PALL Life Science).

MS–MS was performed on eluted peptides as we have previously described[49]. Tandem mass spectra were searched with Sequest (Thermo Fisher Scientific, v.IseNode in Proteome Discoverer 2.5.0.400). Sequest was set to search the mouse Uniprot database (3 July 2020 version) with 55,650 entries, including common contaminants and green fluorescent protein, Cas9, puromycin and P2A (present in the cell lines) assuming no digestion enzyme, with fragment and parent ion mass tolerances of 0.02 Da and 10.0 p.p.m., respectively. TMTpro was added as a fixed modification on the carboxy and amino termini of peptides. Oxidation of methionine was specified in Sequest as a variable modification. The resulting peptides were filtered to exclude peptides with an isolation interference >30% and p.p.m. error >±3 of the median p.p.m. error of all peptide-spectrum matches. Peptides were further prioritized based on concordance of relative abundance across clones with presence/absence of the associated somatic mutation.

## Dendritic cell vaccination, ELISpot and MHC-I multimer staining

Bone marrow-derived dendritic cells were prepared, activated, loaded with putative neoepitopes and injected intradermally at the base of the tail of healthy C57BL/6 mice, followed by two heterologous boosts, as previously described[45]. A week after the second boost, spleen and peripheral blood were collected for IFN-γ ELISpot and MHC-I:epitope tetramer flow cytometric assays. Red blood cells were first lysed with ACK lysis buffer. IFN-γ ELISpot was performed following the manufacturer's recommendations (ImmunoSpot, Cellular Technology Limited) using 750,000 splenocytes per well. H-2K^b tetramers were custom generated as previously described[45] and used at 1:200 dilution. H-2D^b tetramers were generated using UV-labile monomers (UVX Flex-T, BioLegend) following the manufacturer's recommendations and used at 1:50 dilution. H-2K^b:QAYAFLQHL dextramers were generated using the U-Load Dextramer Kit (Immudex) following the manufacturer's recommendations and used at 1:10 dilution.

## Tissue preparation and flow cytometry

Then 2 min before euthanasia, mice were injected retro-orbitally with anti-CD45 APC-eFluor786 (catalog no. 30-F11, BioLegend, 1:50) to stain intravascular immune cells. Mediastinal lymph nodes and whole tumor-bearing lungs were collected and mechanically dissociated in RPMI-1640 (Corning) with 5% heat-inactivated (HI)-FBS (collection medium). Lungs were placed in digestion buffer containing 500 U ml^−1 of collagenase type IV and 20 μg ml^−1 of DNase (Sigma-Aldrich) in collection medium, lightly minced and digested at 37 °C for 30 min with gentle agitation, and further dissociated with a gentleMACS Octo Dissociator (Miltenyi Biotec) on the tumor_imp1.1 setting and passed through a 100-μm filter. Live/dead staining (Ghost Dye Red 780, Corning, 1:500 dilution) was performed in PBS and surface stains in FACS buffer (1 mM EDTA, 25 mM Hepes and 0.5% HI-FBS in PBS). For assessment of T cell depletion (Extended Data Fig. 2m), the following antibodies were used for surface staining: CD45 BV785 (catalog no. 30-F11, BioLegend, 1:200), CD3 BV421 (catalog no. 17A2, BioLegend, 1:400), CD8a BUV395 (catalog no. 53-6.7, BioLegend, 1:400) and CD4 AF647 (catalog no. RM4-5, BioLegend, 1:400). For analysis of neoantigen-specific T cells (Extended Data Fig. 5e), the following antibodies were used for surface staining: CD8a BUV395 (as above), CD4 BV711 (catalog no. RM4-5, BioLegend, 1:200), CD44 BV785 (catalog no. IM7, BioLegend, 1:200) and GZMB PE-CF594 (catalog no. GB11, BD Biosciences, 1:250); and intracellular staining: TCF1 AF647 (catalog no. C63D9, CST, 1:200). Cells were fixed for 1 h at room temperature in Fixation/Permeabilization Concentrate (Thermo Fisher Scientific) diluted 1:3 in Fixation/Permeabilization diluent (Thermo Fisher Scientific) and washed in permeabilization buffer (Thermo Fisher Scientific). Intracellular staining was performed

in permeabilization buffer overnight at 4 °C. Cells were washed and resuspended in FACS buffer for analysis on a BD LSRFortessa four-laser flow cytometer running BD FACSDiva v.8.0 software. Results were analyzed in FlowJo v.10.4.2. Single lymphocytes were gated first on forward versus side scatter (FSC-A versus SSC-A) and then FSC-A versus FSC-H. Then, live CD8+ T cells were gated on positive CD8α and negative Ghost Red Dye 780 staining. Antigen-specific CD8+ T cells were further gated on CD44 positivity and tetramer/dextramer positivity in two channels (PE/APC). Expression of additional markers was analyzed specifically in this neoantigen-specific CD8+ T cell population.

### Analysis of human MMRd cancer clinical trials

Raw WES reads from Bortolomeazzi et al.[50] and Kwon et al.[35] trials (ClinicalTrial.gov identifiers: NCT02563002 and NCT02589496) were mapped to the reference human genome (GRCh38) using BWA-MEM v.0.7.17-r1188 (ref. [65]). Aligned reads were processed as BAMs following the GATK v.4.1.8.0 Best Practices workflow to remove duplicates and recalibrate base quality scores[66]. Somatic SNVs and indels were detected using the same pipeline and callers described above for mouse tumors. CCF values were estimated as described above.

Raw RNA-seq reads were mapped to the human reference genome (GRCh38) using STAR v.2.7.1a[83]. STAR was run using the same parameters as described above in the mouse analysis. The function Htseq-count from the Python library HTSeq (v.2.0.1)[89] was used to compute read counts for each gene (Ensembl release GRCh38.90), which were normalized to transcripts per million. Neoantigens were predicted and prioritized as described above in the mouse analysis. Clonal and subclonal neoantigens were classified as CCF ≥ 0.75 and <0.5, respectively.

Clinical responses were binned into two groups: OR, including partial and complete responders and NR, including patients with stable and progressive disease. PFS analysis was performed on a combination of both trials[35,50] with the trial study as a covariate. Importantly, there was no significant difference in PFS between patients from the two trials. Cox's regression was performed in R (v.4.0.2) using the package survival (v.3.4-0)[90] with comparisons of patients in the upper versus lower quartiles of each variable tested.

### Statistics and reproducibility

Statistical analyses and plotting were performed in R (v.4.2.1) using built-in functions and ggplot2 (v.3.4.1), beeswarm (v.0.4.0), corrplot (v.0.88), eulerr (v.6.1.0), gplots (v.3.1.3), survival (v.3.4.0), survminer (v.0.4.9) and RColorBrewer (v.1.1.3). To assess statistical significance, Fisher's exact 2 × 2 test was used on categorical variables and two-tailed Wilcoxon's rank-sum test or Student's $t$-test (where the assumption of normality was met) was used on continuous variables. HRs were calculated and compared using Cox's proportional hazards regression. Multiple-comparison corrections were performed using Holm's method. No statistical method was used to predetermine sample size. In preclinical trials of lung and colon models, only those animals with apparent tumors by μCT or colonoscopy were recruited. No other data were excluded from analyses. Preclinical trials were randomized and investigators blinded to allocation during dosing, imaging and quantification. No experiments failed to replicate.

### Reporting summary

Further information on research design is available in the Nature Portfolio Reporting Summary linked to this article.

### Data availability

Raw exome sequencing and RNA-seq data from Bortolomeazzi et al.[50] are available through controlled-access application via the European Genome-Phenome Archive (hosted by the EMBL-EBI and the CRG) under accession no. EGAD00001006165. Raw sequencing data from Kwon et al.[35] were downloaded from the European Nucleotide Archive (ENA) database under primary accession no. PRJEB40416. The sequencing data generated in the present study are available at ENA under primary accession no. PRJEB56609. Raw MS data generated in the present study are available at MassIVE under accession no. MSV000092096. Source data are provided with this paper.

### Code availability

Customized scripts used in the analysis of sequencing data in the present study are available at: https://github.com/cortes-ciriano-lab/MMRd_immunogenicity.

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

## Acknowledgements

This work was supported by the National Cancer Institute's Cancer Center Support Grant (nos. P30-CA14051 and R01 CA233983 to T.J.) and the Howard Hughes Medical Institute (to T.J.) P.M.K.W. was supported by a Damon Runyon Fellowship Award. The funders had no role in the study design, data collection and analysis, decision to publish or preparation of the paper. We thank K. Yee, J. Teixeira, K. Anderson and M. Magendantz for administrative support, and our colleagues in the Jacks laboratory and the broader community at the Koch Institute of MIT for thoughtful discussions and technical advice. We thank the Koch Institute Swanson Biotechnology Center for core support from the BioMicro Center and Histology, Proteomics, and Flow Cytometry facilities, with particular thanks to S. Levine for high-throughput sequencing support, K. Cormier and C. Condon for histology support, A. Koller for help with design and analysis of MS experiments, and M. Griffin, M. Jennings and G. Paradis for flow cytometry support. F.M. and I.C.C. thank EMBL for funding. We are also grateful for collaboration with T. Westerling and Aiforia in developing automated CNNs for lung tumor grade and IHC quantification.

## Author contributions

P.M.K.W. and T.J. conceived the study. P.M.K.W., T.J. and I.C.C. supervised the study. P.M.K.W., O.C.S., H.H., N.J.S., A.M.J., W.M.R., D.A.C., G.C.J. and S.L.S. carried out all aspects of animal care and experimentation. F.M. and I.C.C designed sequencing analysis pipelines and carried out analysis with P.M.K.W. Z.A.E. designed and executed the neoantigen prediction pipeline. All other data analysis was carried out by P.M.K.W. A.B. provided conceptual and technical guidance in sequencing analysis. S.N. designed dual sgRNA lentiviral constructs. D.Z., M.C.B., A.J., J.J.P. and A.M.C. produced and validated critical reagents. R.T.B. provided validation of ground truth for training the Aiforia lung tumor grading CNN. P.M.K.W., I.C.C. and T.J. wrote the paper, with feedback from all authors.

## FundingInformation

## Competing interests

T.J. is a member of the Board of Directors of Amgen and Thermo Fisher Scientific, and a co-founder of Dragonfly Therapeutics and T2 Biosystems. He serves on the Scientific Advisory Board of Dragonfly Therapeutics, SQZ Biotech and Skyhawk Therapeutics, and is the President of Break Through Cancer. None of these affiliations represents a conflict of interest with respect to the design or execution of the present study or interpretation of data presented in this manuscript. The Jacks laboratory also currently receives funding from the Johnson & Johnson Lung Cancer Initiative and the Lustgarten Foundation for Pancreatic Cancer Research, but this did not support the research described in the present study. All other authors declare no competing interests.

## Additional information

**Extended data** is available for this paper at https://doi.org/10.1038/s41588-023-01499-4.

**Correspondence and requests for materials** should be addressed to Peter M. K. Westcott, Isidro Cortes-Ciriano or Tyler Jacks.

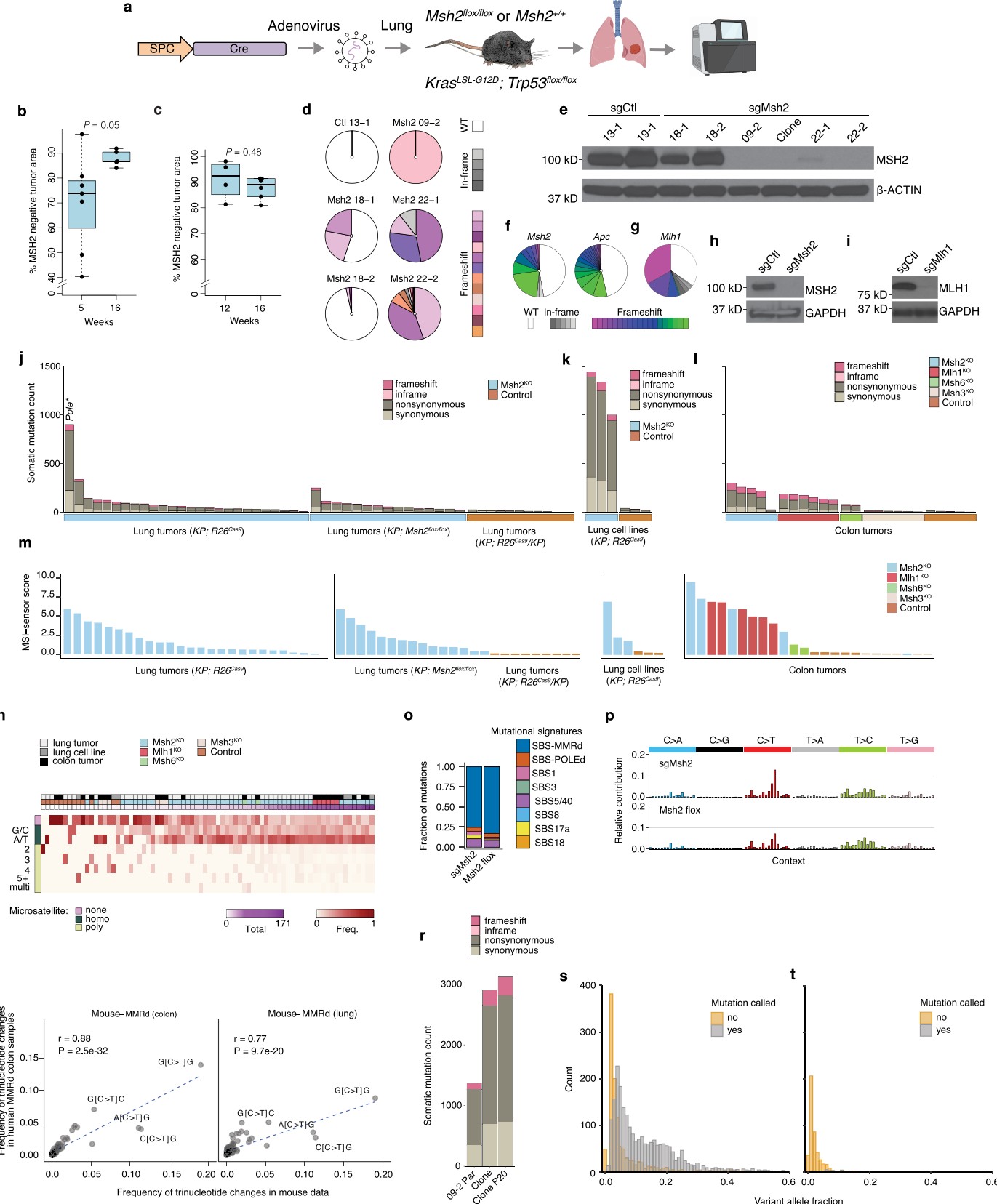

**Extended Data Fig. 1 | See next page for caption.**

**Extended Data Fig. 1 | Validation of *in vivo* DNA mismatch repair gene knockout. (a)** Schematic of *Kras*[LSL-G12D]; *Trp53*[flox/flox]; *Msh2*[floxl/flox] (KPM) lung tumor model. **(b-c)** Percent lung tumor area negative for MSH2 by IHC at 5- and 16-weeks post-initiation of *Kras*[LSL-G12D]; *Trp53*[flox/flox]; *R26*[LSL-Cas9] (KPC) animals with sgMsh2 lentivirus (N = 7 5-week and 5 16-week animals) **(b)** and 12- and 16-weeks post-initiation of KPM animals (N = 4 12-week and 6 16-week animals) **(c)**. **(d)** Deep sequencing of sgMsh2-targeted locus in cell lines derived from 16–20-week sgCtl- and sgMsh2-targeted lung tumors. WT = wild-type. **(e)** Western blot of MSH2 expression in lung tumor cell lines, experimentally replicated three times. Clone = 09-2 single cell clone. **(f-g)** Deep sequencing of *Msh2*, *Apc* **(f)** and *Mlh1* **(g)** loci targeted in autochthonous colon tumors, representative of 5 sgMsh2- and 6 sgMlh1-targeted tumors. **(h-i)** Western blots of MSH2 **(h)** and MLH1 **(i)** expression in one organoid line each derived from sgCtl-, sgMsh2-, and sgMlh1-targeted colon tumors, experimentally replicated twice. **(j-l)** Total somatic single nucleotide variants (SNVs) and insertions/deletions (indels) within exome of autochthonous lung tumors **(j)**, cell lines **(k)**, and autochthonous colon tumors **(l)**. *Pole** = *Pole* S415R mutation. **(m)** MSIsensor-pro scores for tumors in (j-l). **(n)** Frequency of indels across DNA microsatellite contexts, including exonic and intronic mutations. Samples were ordered by total indels. Homo = homopolymer repeats ≥ 4 bases, 2-5 + = microsatellites with motifs of 2-5+ bases, and multi = microsatellites with multiple repetitive motifs. **(o-p)** COSMIC mutational signature decomposition of 15 KPM and 26 KPC (sgMsh2) lung tumors **(o)** based on frequencies of the 96 possible SNVs classified by trinucleotide context **(p)**. SBS_MRD = DNA mismatch repair deficiency (MMRd) signature. **(q)** Pearson correlation of the fraction of mutations detected in each trinucleotide context between human and mouse MMRd datasets. **(r)** Total somatic SNVs/indels in exome of 09-2 lung tumor cell line and early- and 20-passage 09-2 single cell clone. **(s-t)** Total mutations in early-passage clone that are also supported in sequencing reads of 09-2 parental line **(s)** and an unrelated MMRp control line, 13-1 **(t)**, with variant allele fraction on x-axis. Grey = official mutation calls; gold = not called. Significance in (b-c) was assessed by Wilcoxon Rank Sum test.

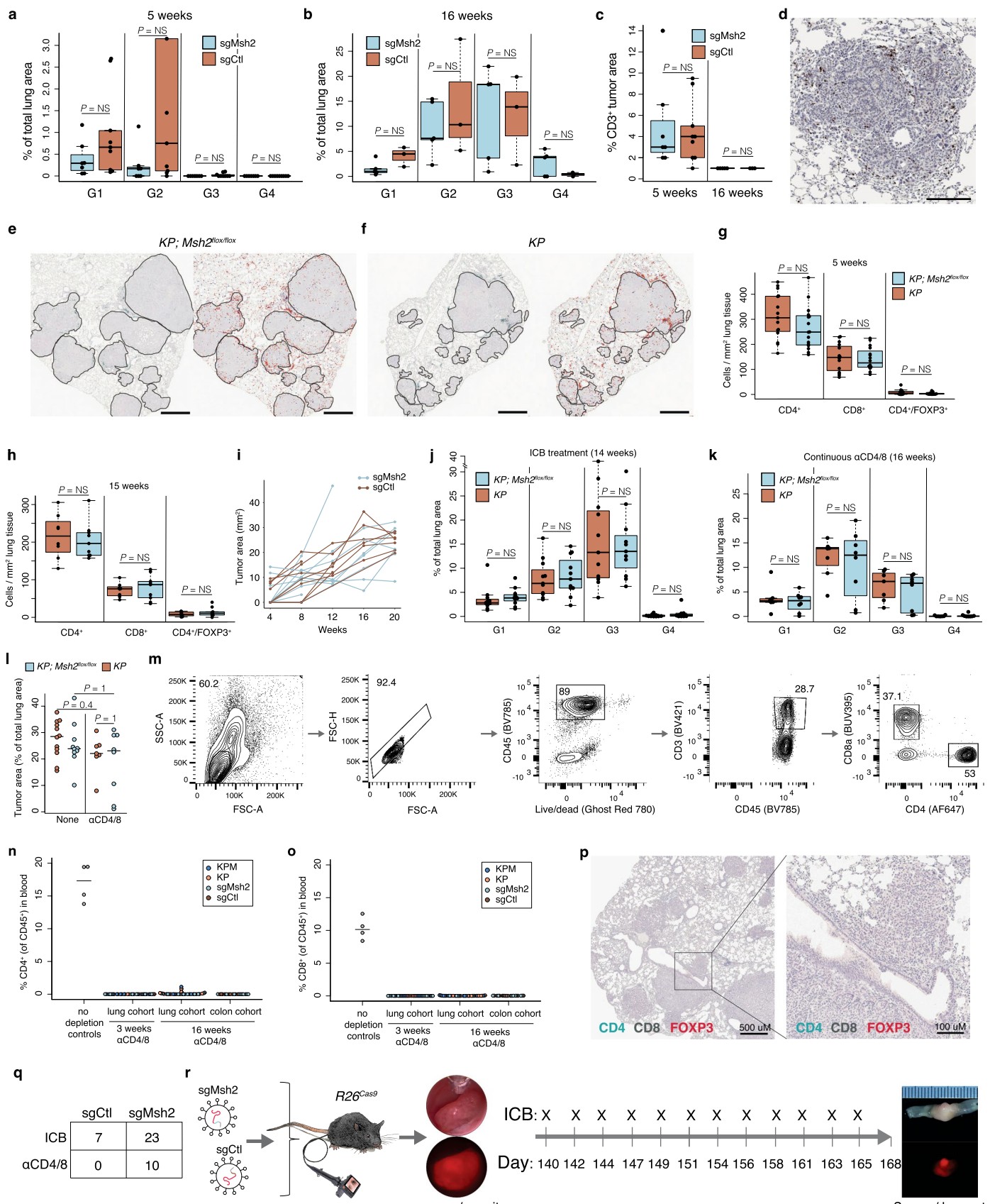

**Extended Data Fig. 2 | See next page for caption.**

**Extended Data Fig. 2 | Tumor kinetics and immunogenicity are unaffected by MMRd. (a-b)** Percent lung area occupied by tumor grades 1-4 (G1-4) in sgMsh2- and sgCtl-targeted KPC mice at 5- **(a)** and 16-weeks **(b)** post-initiation, representative of 7 and 9 5-week and 5 and 3 16-week animals, respectively. **(c-d)** Average lung tumor area of animals in (a-b) positive for CD3 (T cells) by immunohistochemistry (IHC) **(c)**, with representative CD3 IHC of an sgMsh2-targeted lung tumor **(d)**. Scale bar = 200 μM. **(e-h)** IHC staining and Aiforia convoluted neural network (CNN) quantification of KPM and KP tumor-bearing lungs of animals in Fig. 2a,b. Stained 15-week KPM **(e)** and KP **(f)** tumor-bearing lungs (left panel) with Aiforia CNN annotations (right panel) for CD4⁺ (red), CD8⁺ (blue), and CD4⁺FOXP3⁺ Tregs (yellow), with quantification across whole lungs in KPM and KP models at 5- **(g)** and 15-weeks **(h)** post-initiation. Tumors in (e-f) are outlined in black; scale bar = 1 mm. **(i)** Change in focal colon tumor area by longitudinal colonoscopy. N = 7 sgMsh2- and 7 sgCtl-targeted animals with 10 and 9 tumors, respectively. **(j-l)** Percent lung area occupied by G1-4 tumors in KPM and KP mice (from Fig. 2i) after 4 weeks of ICB treatment **(j)**, at 16-weeks post-initiation with continuous αCD4/8 treatment **(k)**, and overall lung tumor burden with no treatment versus continuous αCD4/8 at 14- and 16-weeks post-initiation, respectively **(l)**. **(m-o)** Flow cytometric analysis of CD4⁺ and CD8⁺ T cells in peripheral blood of experimental animals following αCD4/8 treatment. Gating strategy **(m)**. Percent of CD4⁺ **(n)** and CD8⁺ **(o)** T cells relative to total CD45⁺ cells. **(p)** CD4, CD8, and FOXP3 IHC of 16-week tumor-bearing lungs from animals treated continuously with αCD4/8. Animals in (n-p) are the same as αCD4/8-treated animals in (k) and (q). **(q-r)** Colon preclinical trial arms **(q)** and schematic **(r)**. Boxplots display median, interquartile range (box bounds), whiskers extending to most extreme points (≤1.5X interquartile range), and all datapoints. Significance in (a-c, g-h, j-l) was assessed by Wilcoxon Rank Sum test with multiple test corrections.

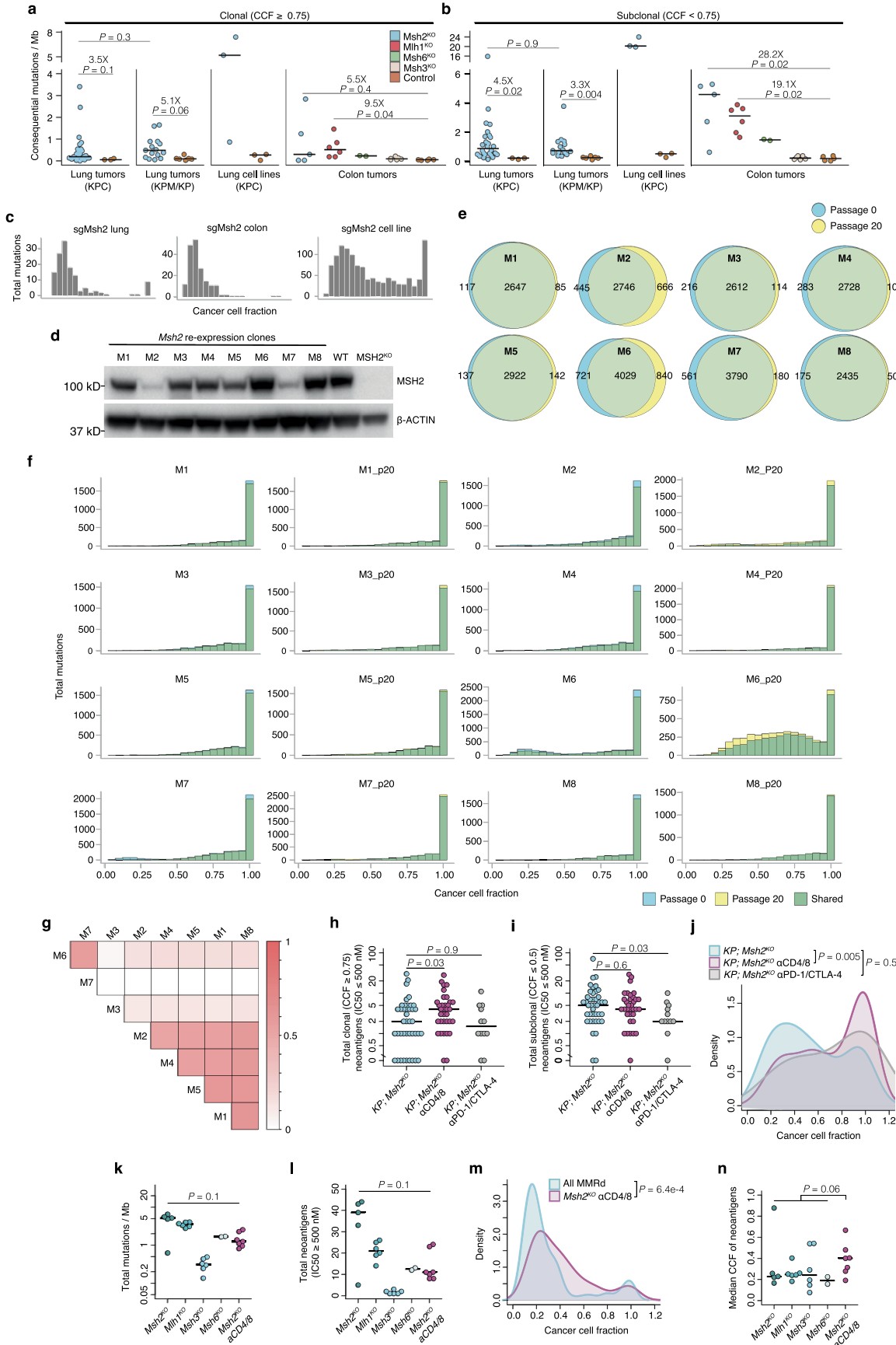

**Extended Data Fig. 3 | See next page for caption.**

**Extended Data Fig. 3 | Immunoediting exacerbates intratumoral heterogeneity by pruning clonal but not subclonal neoantigens. (a-b)** Total consequential mutations (nonsynonymous SNVs and indels) per megabase (Mb) DNA for autochthonous lung tumors and cell lines and autochthonous colon tumors, separated by clonal (cancer cell fraction (CCF) ≥ 0.75) **(a)** and subclonal (CCF ≤ 0.75) **(b)**, with fold-change shown for each comparison. **(c)** Histograms of total mutations by cancer cell fraction in a representative sgMsh2-targeted lung tumor, cell line, and colon tumor from Fig. 3a,b. **(d)** Western blot of MSH2 expression in single cell clones with *Msh2* re-expression, after 20 passages with puromycin selection, experimentally replicated three times. WT = sgCtl-targeted cell line; MSH2$^{KO}$ = parental sgMsh2-targeted cell line (09-2). **(e)** Venn diagrams of mutation overlap between M1-8 clones sequenced at early passage (called passage 0 for convenience) and 20 passages later. **(f)** Histograms of total mutations by cancer cell fraction in M1-8 clones at passage 0 and 20. **(g)** Pairwise

intersection map of mutations across M1-8 clones. Scale represents fraction of total mutations shared between each pair. **(h-i)** Total clonal (CCF ≥ 0.75) **(h)** and subclonal (CCF ≤ 0.5) **(i)** predicted neoantigens in lung tumors from Fig. 3g–j. **(j)** Distribution of cancer cell fraction estimates from Fig. 3i with sgMsh2-targeted *Pole* S415R mutant lung tumor removed. **(k-l)** Total mutations / Mb **(k)** and predicted neoantigens **(l)** in 16–20-week autochthonous MMRd colon tumors from animals with no treatment (blue shades, N = 5 *Msh2$^{KO}$*, 6 *Mlh1$^{KO}$*, 6 *Msh3$^{KO}$*, 2 *Msh6$^{KO}$*) and continuous antibody-mediated T cell depletion (αCD4/8, magenta, N = 7 *Msh2$^{KO}$*). **(m-n)** CCF distribution **(m)** and per tumor median **(n)** of all SNV-derived neoantigens (no expression filter) in colon tumors from (j-k). Significance and smoothing in (i, l) were assessed by two-sided Kolmogorov-Smirnov test and Gaussian kernel density estimation, respectively. Significance in (a-b, h-i, k-l, n) was assessed by Wilcoxon Rank Sum test, with Holm correction for five comparisons in (a-b). *P* values in (h-j) are uncorrected.

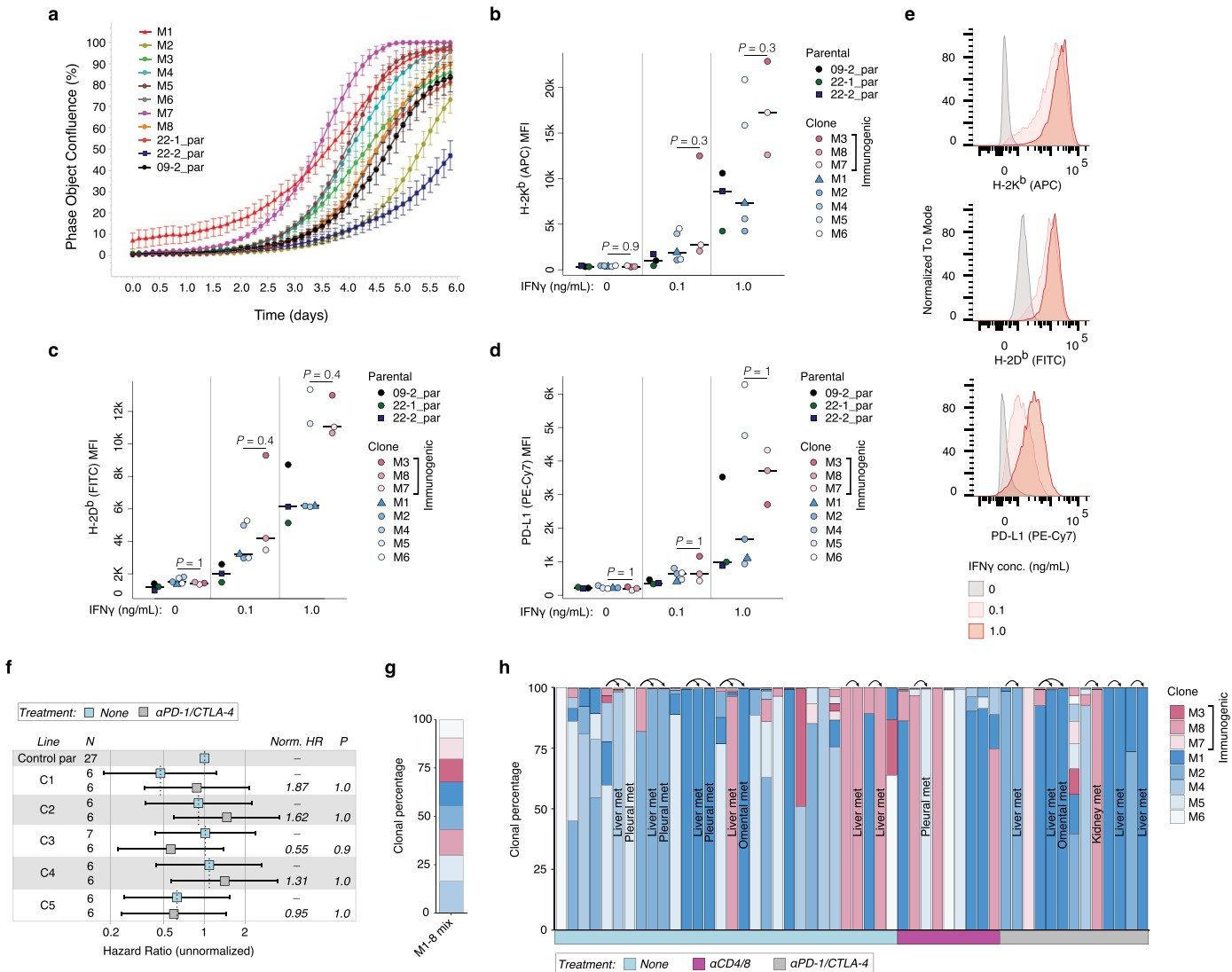

**Extended Data Fig. 4 | Clones re-expressing *Msh2* grow similarly *in vitro* and are IFNγ responsive. (a)** *In vitro* growth kinetics of parental MSH2 knockout cell lines (09-2, 22-1, 22-2) and *Msh2* re-expressing clones generated from 09-2 (M1-8), measured by live cell imaging with an IncuCyte S3. Bars represent standard deviation of eight replicates (wells). **(b-e)** Flow cytometric analysis of surface expression of MHC-I alleles H-2K$^b$ **(b)** and H-2D$^b$ **(c)** and IFNγ-response gene PD-L1 **(d)** in cell lines from (a) following overnight stimulation with 0, 0.1, and 1.0 ng/mL IFNγ. MFI = mean fluorescence intensity. **(e)** Representative histograms of H-2K$^b$, H-2D$^b$ and PD-L1 expression in a clone (M3) from the experiment in (b-d). **(f)** Survival Hazard Ratios (HR) of syngeneic mice orthotopically transplanted via intratracheal instillation with clones (C1-5) derived from parental sgCtl-

targeted line 13-1 (*Msh2* WT), with and without ICB treatment. *N* = number of animals, *P* = P-value, *Norm.HR* = normalized HR. Bars represent upper and lower 95% confidence intervals. **(g-h)** Estimation of clonal frequencies of M1-8 clones in equimolar mixture of DNA **(g)** and lung tumors and associated metastases (denoted by arrows) from animals transplanted with an equal mixture of all clones and receiving no treatment, continuous αCD4/8, or ICB **(h)**, as determined by targeted deep amplicon sequencing of 4 private SNVs per clone. ICB treatment in (f, h) was started 2-weeks post-transplant and continued for 4 weeks. Significance in (b-d) was assessed by Wilcoxon Rank Sum test with correction for 3 tests. Significance in (f) was assessed by Cox proportional hazards regression with Holm correction for 12 tests (other 7 tests shown in Fig. 4b).

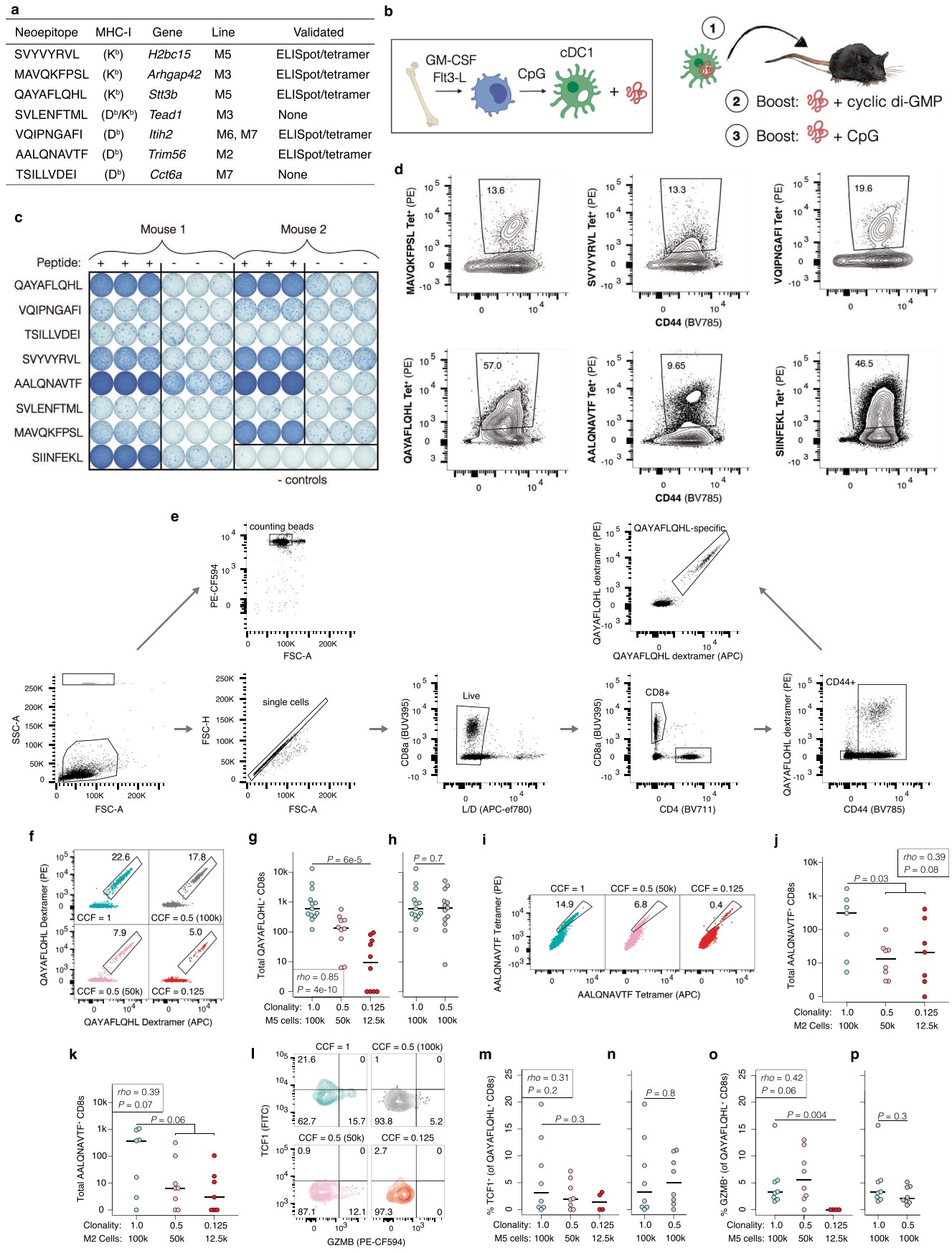

**Extended Data Fig. 5 | See next page for caption.**

**Extended Data Fig. 5 | Identification and validation of *bona fide* MMRd-derived neoantigens. (a)** Somatic mutation-derived epitopes (neoepitopes) found to be presented on surface MHC-I (H2-K$^b$ and H2-D$^b$) in M1-8 clones by Tandem Mass Tag Mass Spectrometry. Immunogenicity was validated by IFNγ ELISpot and MHC-I tetramer staining. **(b)** Vaccine regimen to assess in vivo immunogenicity of neoepitopes. Conventional cross-presenting dendritic cells (cDC1s) were in vitro differentiated from bone marrow, activated, loaded with neoepitope peptide and injected intradermally into mice, which were boosted with two different adjuvants delivered with peptide over the course of the experiment (see Methods). **(c-d)** IFNγ ELISpot **(c)** and flow cytometric staining with neoepitope-loaded MHC-I tetramers **(d)** of splenocytes isolated from mice vaccinated with the indicated peptides. SIINFEKL = immunogenic peptide positive control; "- controls" = no peptide vaccination. **(e)** Flow cytometric gating strategy for analysis of neoantigen-specific CD8$^+$ T cells in Fig. 5. **(f-h)** Representative flow plot **(f)** and total QAYAFLQHL-specific CD8$^+$ T cells

**(g-h)** in lungs from animals in Fig. 5 as determined by MHC-I dextramer staining in two channels (PE and APC). **(i-k)** Flow cytometric analyses of M2 clone neoantigen (AALQNAVTF)-specific T cells isolated from mLNs and lungs of syngeneic mice intratracheally transplanted with M2 at CCF = 1 (N = 7), 0.5 (N = 8), and 0.125 (N = 7), and treated with ICB for 2 weeks starting 2-weeks post-transplant. Representative flow plot **(i)** and total AALQNAVTF-specific CD8$^+$ T cells in mLNs **(j)** and lungs **(k)** as determined by MHC-I tetramer staining in two channels (PE and APC). **(l-p)** Representative flow plot **(l)** and quantification of percent of QAYAFLQHL-specific CD8$^+$ T cells positive for TCF1 **(m-n)** and GZMB **(o-p)** in lungs from animals in Fig. 5. Significance in (g, j-k, m, o) was assessed by both Spearman Rank Correlation with a numeric x-axis (CCF) and Wilcoxon Rank Sum test (CCF = 1 versus 0.125 groups). Significance in (h, n, p) was assessed by Wilcoxon Rank Sum test. Samples with < 10 QAYAFLQHL-specific CD8$^+$ T cells were excluded from analysis in (m-p).

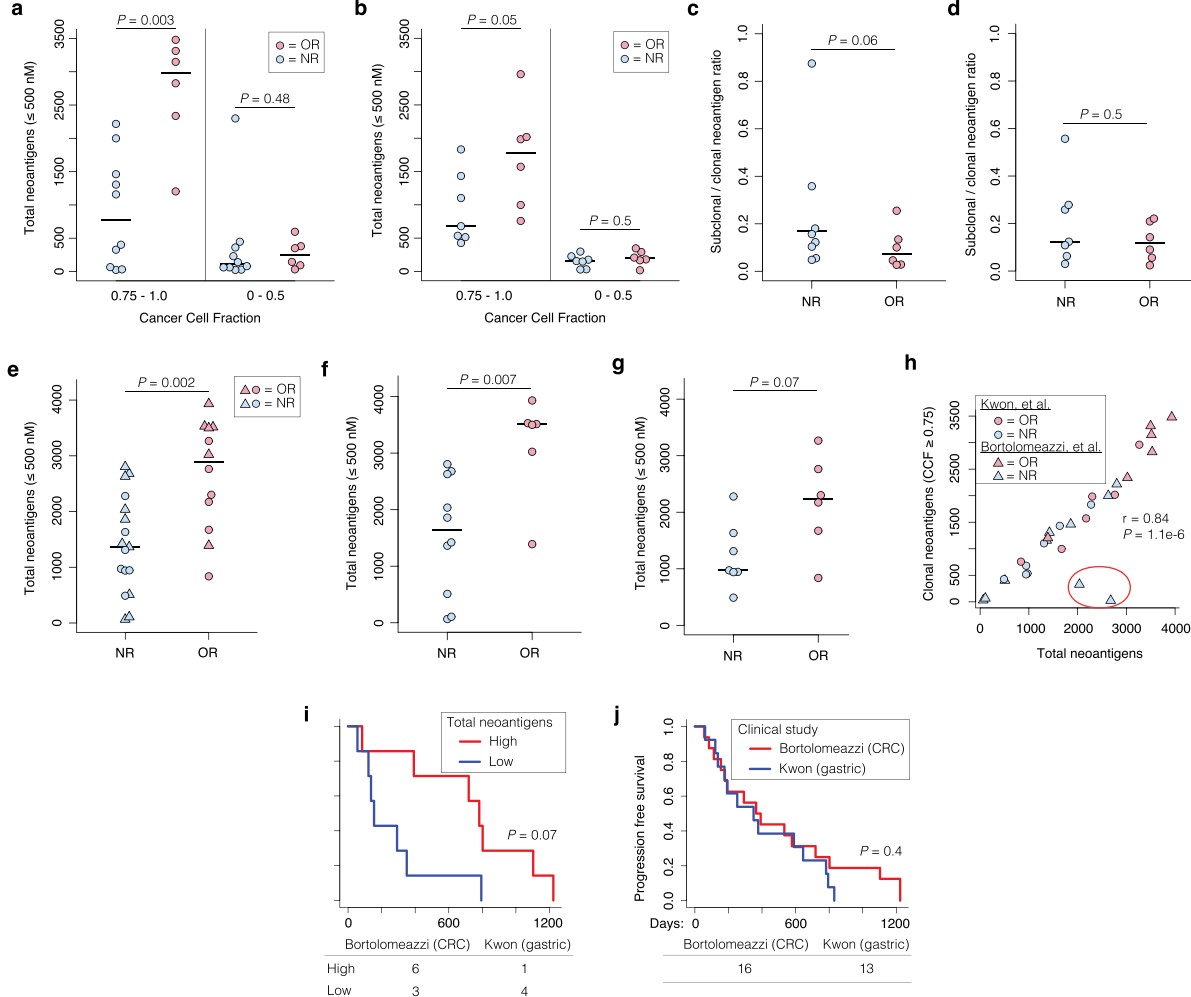

**Extended Data Fig. 6 | Overall tumor neoantigen burden is imperfectly correlated with clonal burden and ICB response in human MMRd cancer.** (a-d) Total clonal (CCF = 0.75 to 1.0) and subclonal (CCF = 0 to 0.5) tumor neoantigen burden (TNB) (a-b) and ITH index (subclonal to clonal neoantigen ratio) (c-d) in patients with objective response (OR) versus nonresponse (NR) in separated analyses of Bortolomeazzi trial[50] (a,c) and Kwon trial[35] (b,d). (e-g) TNB, regardless of clonality, in combined analysis (e) and separated analyses of

Bortolomeazzi (f) and Kwon (g) trials. (h) Pearson correlation of overall versus clonal TNB across both studies, with correlation outliers circled in red. (i-j) Progression free survival of patients separated by upper versus lower quartiles of overall TNB (i) and clinical study (j). Number of patients from each study is indicated under plots. Significance in (a-g) was assessed by Wilcoxon Rank Sum test, and in (i-j) was assessed by Cox proportional hazards regression, with clinical trial study as a covariate in (i).

# Reporting Summary

## Statistics

For all statistical analyses, confirm that the following items are present in the figure legend, table legend, main text, or Methods section.

| n/a | Confirmed | |
|---|---|---|
| ☐ | ☒ | The exact sample size (*n*) for each experimental group/condition, given as a discrete number and unit of measurement |
| ☐ | ☒ | A statement on whether measurements were taken from distinct samples or whether the same sample was measured repeatedly |
| ☐ | ☒ | The statistical test(s) used AND whether they are one- or two-sided *Only common tests should be described solely by name; describe more complex techniques in the Methods section.* |
| ☐ | ☒ | A description of all covariates tested |
| ☐ | ☒ | A description of any assumptions or corrections, such as tests of normality and adjustment for multiple comparisons |
| ☐ | ☒ | A full description of the statistical parameters including central tendency (e.g. means) or other basic estimates (e.g. regression coefficient) AND variation (e.g. standard deviation) or associated estimates of uncertainty (e.g. confidence intervals) |
| ☐ | ☒ | For null hypothesis testing, the test statistic (e.g. *F*, *t*, *r*) with confidence intervals, effect sizes, degrees of freedom and *P* value noted *Give P values as exact values whenever suitable.* |
| ☒ | ☐ | For Bayesian analysis, information on the choice of priors and Markov chain Monte Carlo settings |
| ☒ | ☐ | For hierarchical and complex designs, identification of the appropriate level for tests and full reporting of outcomes |
| ☒ | ☐ | Estimates of effect sizes (e.g. Cohen's *d*, Pearson's *r*), indicating how they were calculated |

*Our web collection on statistics for biologists contains articles on many of the points above.*

## Software and code

Policy information about availability of computer code

| Data collection | μCT data were captured using a GE eXplore CT 120 system. Endoscope images were captured using a Karl Storz AIDA HD capture system. Flow cytometry data were collected on a BD LSRFortessa using BD FACSDiva v8.0 software. |
|---|---|
| Data analysis | A custom pipeline was constructed for somatic SNV and indel calling using whole-exome sequencing data employing the following published algorithms: BWA-MEM v0.7.17-r1188, Genome Analysis Toolkit (GATK) v4.1.8.0, MuSE v1.0rc, VarDict v1.8.2, Strelka2 v2.9.2, and SomaticCombiner v1.03. Microsatellite contexts of mutations were annotated using SciRoKo v3.4. Copy number aberrations were detected using FreeBayes v1.3 and PureCN v1.16.0, and cancer cell fraction was estimated using the package cDriver in R v4.0.2. Mutational signature analysis was performed using the R (v4.0.2) package MutationalPatterns71 (v3.2.0) and COSMIC Mutational Signatures catalogue version 3. We used the function fit_to_signatures with default parameter values to estimate the contribution of each mutational process to the observed mutational spectrum in sample group. For fitting the signatures, we only used the mutational processes known to be operative in human colon and/or lung cancer, except for tobacco smoking: SBS1, SBS5, SBS6, SBS10a, SBS10b, SBS14, SBS15, SBS17a, SBS17b, SBS18, SBS21, SBS26, SBS28, SBS37, SBS40, and SBS44. For visualization purposes, we collapsed the contribution of mutational signatures associated with defective DNA mismatch repair (SBS6, SBS14, SBS15, SBS21, SBS26 and SBS44; labelled as MMRd in the figures) and POLE deficiency (SBS10a, SBS10b, SBS28, SBS17b; labelled as POLE in the figures). MSI status of samples was calculated using the software MSIsensor-pro (v1.2.0) with matched normal tails as controls. Mouse tumor purity estimates based on Trp53-flox allele recombination and Msh2-flox allele recombination efficiency were calculated using SAMtools v1.10 and a custom script in R v4.0.2.<br><br>A custom pipeline was constructed for neoantigen prediction employing the following published algorithms: Ensembl Variant Effect Predictor (VEP) v99, pVACtools v1.5.7, NetMHC-4.0, NetMHCpan-4.0, SMM v1.0, SMMPMBEC v1.0, and custom scripts in Python v2.7.13 as previously described (Westcott, PMK, et al., Nature Cancer, 2021 in press). Expression was calculated from RNA sequencing data using STAR v2.7.1a and Picard v2.23.4, and allele-specific expression of mutations associated with predicted neoantigens in the RNA sequencing BAMs was calculated using custom scripts in Python v2.7.13. |

Clonal deconvolution of targeted amplicon sequencing data was performed using the following published algorithms: BWA-MEM v0.7.17-r1188, bcftools v1.10.2, and custom scripts in R v4.0.2 and Python v2.7.13. or CleanPlex UMI libraries, fgbio (v2.0.1) was used to extract UMIs (ExtractUmisFromBam) and call consensus reads (GroupReadsByUmi, CallMolecularConsensusReads).

Tumor burden and grade of whole lung H&E sections and triple IHC staining (CD8, CD4, FOXP3) of histological sections were quantified using custom convoluted neural network developed with Aiforia's cloud-based image analysis platform. This is a commercial platform with proprietary technology and therefore did not generate any code. An interactive example of algorithm functionality can be provided free-of-charge upon request at https://www.aiforia.com. MSH2 staining of lung tumors was quantified by manual annotation with QuPath v0.1.2. Quantification of CD3 staining in tumors of lung histological sections was performed with Aperio ImageScope v12.1.

Longitudinal change in lung tumor burden as measured by μCT was calculated using a custom MATLAB (MathWorks) script, as previously described (Tammela, T, et al., Nature 2017). Colon tumor area was calculated from longitudinal endoscope and endpoint stereoscope images using ImageJ v2.1.0/1.53c.

Flow cytometry results were analyzed in FlowJo v10.4.2.

Tandem mass spectra were searched with Sequest (Thermo Fisher Scientific, San Jose, CA, USA; version IseNode in Proteome Discoverer 2.5.0.400). Sequest was set up to search a mouse uniprot database (database v.July 3, 2020; 55650 entries containing common contaminants and the proteins GFP, Cas9, Puromycin, and P2A (present in the cell lines) assuming no digestion enzyme (unspecific).

Statistical analyses and figure generation were performed in R (v4.2.1) using built in functions and ggplot2 (v3.4.1), beeswarm (v0.4.0), corrplot (v0.88), eulerr (v6.1.0), gplots (v3.1.3), survival (v3.4.0), survminer (v0.4.9), and RColorBrewer (v1.1.3).

All custom code used in this study are available from the authors upon request.

For manuscripts utilizing custom algorithms or software that are central to the research but not yet described in published literature, software must be made available to editors and reviewers. We strongly encourage code deposition in a community repository (e.g. GitHub). See the Nature Portfolio guidelines for submitting code & software for further information.

# Data

Policy information about availability of data

All manuscripts must include a data availability statement. This statement should provide the following information, where applicable:
- Accession codes, unique identifiers, or web links for publicly available datasets
- A description of any restrictions on data availability
- For clinical datasets or third party data, please ensure that the statement adheres to our policy

Raw exome sequencing and RNA-seq data from Bortolomeazzi et al. (48) are available through controlled-access application via the European Genome-Phenome Archive (EGA, hosted by the EMBL-EBI and the CRG) under the accession number EGAD00001006165. Raw sequencing data from Kwon et al. (35) were downloaded from the European Nucleotide Archive (ENA) database under primary accession number PRJEB40416. The sequencing data generated in this study are available at ENA under primary accession number PRJEB56609. Raw mass spectrometry data generated in this study are available at MassIVE under accession number MSV000092096.

# Field-specific reporting

Please select the one below that is the best fit for your research. If you are not sure, read the appropriate sections before making your selection.

☒ Life sciences        ☐ Behavioural & social sciences        ☐ Ecological, evolutionary & environmental sciences

For a reference copy of the document with all sections, see nature.com/documents/nr-reporting-summary-flat.pdf

# Life sciences study design

All studies must disclose on these points even when the disclosure is negative.

| Sample size | No sample size power calculations were performed. Numbers were chosen based on a combination of past experience with similar studies and practicality (e.g., available animals, funds), with the aim of including at least 10 animals in each group. In the orthotopic transplant studies, this was not possible due to the large number of groups and treatments and the prolonged nature of survival studies. Therefore, we aimed to include at least 5 animals within each group in these studies. Orthotopic transplant survival studies were also performed on a rolling basis as mice became available, given the large number of animals used in these studies. Preclinical treatment studies were performed across two independent cohorts in both the lung and colon models with the aim of validating consistency of results. |
|---|---|
| Data exclusions | Only mice with discrete lung tumors by μCT at 10 weeks and mice with clear mScarlet positive (red fluorescent) colon tumors by colonoscopy at 20 weeks were recruited into treatment arms. No other data were excluded. |
| Replication | All in vivo experiments were repeated at least 2-3 times (as described in Sample size and Data exclusions above) to verify reproducibility. In vitro growth kinetics and IFN gamma sensitivity experiments with cell lines were performed twice. No experiments presented in this manuscript failed to replicate. Whole-exome (WES), targeted amplicon, and RNA sequencing were performed once per animal in two separate batches for RNA-seq and targeted amplicon sequencing and four separate batches for WES. No significant batch effects were observed. |
| Randomization | Randomization was performed in assigning mice to treatment arms in preclinical trials. μCT was performed at 10 weeks and colonoscopy at 20 weeks and mice without discrete tumors excluded. Mice of each sex (approximately equal numbers) were then randomly and evenly (as |

possible) assigned across treatment arms using the Sample function in R v4.0.2. Randomization was not appropriate for other experiments presented in this manuscript as mice were age-, litter-, and sex-matched for consistency, and downstream treatment, sampling, and endpoints were identical across all animals. Only male recipient mice were used in orthotopic transplant experiments, as the tumor cell lines transplanted all carry the Y chromosome.

Blinding | Investigators were blinded to genotypes and treatment groups when dosing during preclinical trials, performing μCT and colonoscopy, quantifying tumor burden from colonoscopy images, stereoscope images, and H&E histological lung sections, and quantifying CD3 infiltration by immunohistochemistry. Quantification of μCT data, H&E tumor burden in the KPM cohort and preclinial trial, and infiltration of CD4, CD8, and Regulatory T cells was performed algorithmically and thus was intrinsically blinded. All other analyses where statistical comparisons were made between different groups were performed blinded.

# Reporting for specific materials, systems and methods

We require information from authors about some types of materials, experimental systems and methods used in many studies. Here, indicate whether each material, system or method listed is relevant to your study. If you are not sure if a list item applies to your research, read the appropriate section before selecting a response.

## Materials & experimental systems

| n/a | Involved in the study |
|-----|----------------------|
| ☐ | ☒ Antibodies |
| ☐ | ☒ Eukaryotic cell lines |
| ☒ | ☐ Palaeontology and archaeology |
| ☐ | ☒ Animals and other organisms |
| ☒ | ☐ Human research participants |
| ☒ | ☐ Clinical data |
| ☒ | ☐ Dual use research of concern |

## Methods

| n/a | Involved in the study |
|-----|----------------------|
| ☒ | ☐ ChIP-seq |
| ☐ | ☒ Flow cytometry |
| ☒ | ☐ MRI-based neuroimaging |

## Antibodies

Antibodies used |
In vivo dosing:
CD4 (GK1.5, BioXCell Cat#: BE0003-1) (see Methods for in vivo dosing)
CD8 (2.43, BioXCell Cat#: BP0061) (see Methods for in vivo dosing)
PD-1 (29F.1A12, BioXCell Cat#: BE0273) (see Methods for in vivo dosing)
CTLA-4 (9H10, BioXCell Cat#: BE0131) (see Methods for in vivo dosing)

Westerns:
MSH2 (D24B5, Cell Signaling Technology Cat#: 2017) 1:1000
MLH1 (EPR3894, Abcam Cat#: ab92312) 1:1000
GAPDH (6C5, Santa Cruz Cat#: sc-32233) 1:5000
β-ACTIN (13E5, Cell Signaling Technology Cat#: 4970) 1:5000

Immunohistochemistry:
MSH2 (polyclonal, Abcam Cat#: ab70270) 1:1000
CD8a (EPR21769, Abcam Cat#: ab217344) 1:1000
FOXP3 (FJK-16s, eBioscience Cat#: 14-5773-82) 1:125
CD4 (EPR19514, Abcam Cat#: ab183685) 1:400
CD3 (polyclonal, Abcam Cat#: ab5690) 1:1000
HRP anti-Rabbit IgG (Vector Cat#: MP-7401) 4 drops
AP anti-Rabbit IgG (Vector Cat#: MP-5401) 4 drops

Flow cytometry:
H-2Kb APC (AF6-88.5.5.3, Thermo Fisher Cat#: 17-5958-82) 1:200
H-2Db FITC (28-14-8, Thermo Fisher Cat#: 11-5999-82) 1:200
PD-L1 PE-Cy7 (10F.9G2, BioLegend Cat#: 124313) 1:200
CD8a BUV395 (53-6.7, BioLegend Cat#: 563786) 1:400
CD3 BV421 (17A2, BioLegend, Cat#: 100227) 1:400
CD44 BV785 (IM7, BioLegend Cat#: 103057) 1:200
CD4 AF647 (RM4-5, BioLegend Cat#: 100530) 1:400
CD4 BV711 (RM4-5, BioLegend Cat#: 100549) 1:200
GZMB PE-CF594 (GB11, BD Biosciences Cat#: 562462) 1:250
TCF1 AF647 (C63D9, CST Cat#: 37636S) 1:200
CD45 BV785 (30-F11, BioLegend Cat#: 103149 ) 1:200
CD45 APC-eFluor 780 (30-F11, eBioscience, Cat#: 47-0451-82) 1:50

QAYAFLQHL H-2Kb tetramer PE/APC (custom purified disulfide stabilized H-2Kb) 1:200
QAYAFLQHL H-2Kb dextramer PE/APC (easYmer H-2Kb U-Load Dextramer, Immudex, cat#: U-LX42) 1:10
MAVQKFPSL H-2Kb tetramer PE/APC (custom purified disulfide stabilized H-2Kb) 1:200
SVYVYRVL H-2Kb tetramer PE/APC (custom purified disulfide stabilized H-2Kb) 1:200
SVLENFTML H-2Kb tetramer PE/APC (custom purified disulfide stabilized H-2Kb) 1:200
SVLENFTML H-2Db tetramer PE/APC (Flex-T UVX H-2Db, BioLegend, cat#: custom/pre-catalog) 1:50

AALQNAVTF H-2Db tetramer PE/APC (Flex-T UVX H-2Db, BioLegend, cat#: custom/pre-catalog) 1:50
VQIPNGAFI H-2Db tetramer PE/APC (Flex-T UVX H-2Db, BioLegend, cat#: custom/pre-catalog) 1:50
TSILLVDEI H-2Db tetramer PE/APC (Flex-T UVX H-2Db, BioLegend, cat#: custom/pre-catalog) 1:50
SIINFEKL tetramer PE/APC (custom purified disulfide stabilized H-2Kb) 1:200

| Validation | Abcam antibodies against CD3, CD8, CD4, and FOXP3 and optimal dilutions used for IHC were validated by confirming expected staining patterns of T cells in positive control spleen and lymphnode tissues, with minimal background in non-lymphoid tissue. The Abcam MSH2 antibody for IHC was validated using sgMsh2-targeted lungs and colons (containing knockout tumors) and control tissue (normal lung, colon, and spleen, and sgCtl-targeted tumors), and showed the expected pattern of constitutive expression in normal tissue and control tumors, and specific knockout in most sgMsh2-targeted tumors. Note: this polyclonal antibody did not perform consistently across batches, with some staining non-specifically. Each lot had to be tested separately. Western antibodies against MSH2 and MLH1 were validated using sgRNA-targeted knockout cell lines and positive control cell lines (without specific targeting). anti-CD4 (GK1.5, BioXCell Cat#: BE0003-1) and anti-CD8 (2.43, BioXCell Cat#: BP0061) at the indicated in vivo doses (see Methods) showed nearly complete depletion of CD4 and CD8 T cells in our models by flow cytometry of peripheral blood, spleens, draining LNs, and colons following two doses. On target depletion of PD-1 on T cells following treatment with anti-PD-1 (29F.1A12, BioXCell Cat#: BE0273) was confirmed by flow cytometry of peripheral blood (using a different anti-PD-1 clone, RMP1-30) of tumor-bearing mice every week following treatment initiation in a previously published model (Westcott, PMK, et al., Nature Cancer, 2021). All other antibodies were used following manufacturer recommendations or most commonly described dilutions in the literature. For flow cytometry, specificity of all antibodies was confirmed by comparing against fluorescence minus one (FMO) controls. |
|---|---|

# Eukaryotic cell lines

Policy information about cell lines

| Cell line source(s) | Lung tumor cell lines, single cell clones, and the mouse ES cell line 12A2 used in this study were developed by the authors. HEK-293 cells were ordered from ATCC. |
|---|---|
| Authentication | Authentication of genetic events (MSH2 knockout, MSH2 re-expression) were confirmed during the establishment of lung tumor cell lines as described in the manuscript. Authentication of knock-in alleles in 12A2 was previously confirmed by Southern blot. HEK-293 cells were not authenticated. |
| Mycoplasma contamination | Lung tumor cell lines, single cell clones, and HEK-293 cells used to generate lentivirus have been routinely tested and confirmed negative for mycoplasma contamination. |
| Commonly misidentified lines (See ICLAC register) | None |

# Animals and other organisms

Policy information about studies involving animals; ARRIVE guidelines recommended for reporting animal research

| Laboratory animals | Autochthonous colon cancer experiments were performed in pure C57BL/6 R26-Cas9 mice. Autochthonous lung cancer experiments were performed in mixed C57BL/6; 129/SvJ mice (KP; R26-LSL-Cas9) and pure C57BL/6 mice (KP, KP; Msh2-flox/flox). Mice in autochthonous and orthotopic models were 6-12 weeks and 10-16 weeks of age at initiation, respectively. Approximately equal numbers of male and female mice were used in all studies, with the exception of the orthotopic lung transplantation experiments, where only male albino C57BL/6 hosts chimeric for the male ES cell line 12A2 (KP; R26LSL-Cas9, mixed C57BL/6 and 129/SvJ) were used. Mice were housed in a facility with a 12-hour light/12-hour dark cycle with temperatures within 68-72°F and 30-70% humidity. |
|---|---|
| Wild animals | This study did not involve wild animals. |
| Field-collected samples | This study did not involve field-collected samples. |
| Ethics oversight | All mouse work was approved by the Department for Comparative Medicine (DCM) at MIT and the Institutional Animal Care and Use Committee (IACUC) |

Note that full information on the approval of the study protocol must also be provided in the manuscript.

# Flow Cytometry

## Plots

Confirm that:

☒ The axis labels state the marker and fluorochrome used (e.g. CD4-FITC).

☒ The axis scales are clearly visible. Include numbers along axes only for bottom left plot of group (a 'group' is an analysis of identical markers).

☒ All plots are contour plots with outliers or pseudocolor plots.

☒ A numerical value for number of cells or percentage (with statistics) is provided.

## Methodology

| Sample preparation | Cell lines were grown to 70-90% confluence, trypsinized for 10-15 minutes at 37 degrees, resuspended in media and |
|---|---|

| Sample preparation | pelleted. Cells were then washed in PBS, filtered, and live/dead stained in PBS. Cells were then resuspended in FACS buffer (see Methods) and surface staining performed on ice for 45 minutes. Samples were then run on the flow cytometer. |
| --- | --- |
| Instrument | BD LSR Fortessa Flow Cytometer with 355 nm, 405 nm, 488 nm, and 640 nm excitation lasers. |
| Software | BD FACSDIVA v8.0 was used to collect data. FlowJo v10.4.2 was used to analyze data. |
| Cell population abundance | Populations were not sorted for downstream manipulation in this study. |
| Gating strategy | Cell lines were first gated on FSC-A vs SSC-A, then on FSC-A vs FSC-H for single cells, and then on negative staining for the live/dead ghost dye red 780 (Corning). Mean fluorescent intensity and histograms of H-2Kb (APC), H-2Db (FITC), and PD-L1 (PE-Cy7) were then calculated on this population.

Lung and lymph nodes were first gated on FSC-A vs SSC-A, then on FSC-A vs FSC-H for single cells, and then on negative staining for the live/dead ghost dye red 780. Intravascular cells were excluded by gating for IV CD45 negativity. In T cell depletion experiments, CD4 and CD8a positivity was then used to quantify numbers of these population. In M1-8 clonal mixing transplant experiments, neoantigen-specific T cells were gated by CD8a positivity, CD4 negativity, CD44 positivity, and tetramer/dextramer double positivity (PE and APC). TCF1 and GZMB staining was specifically assessed in this population. |

☒ Tick this box to confirm that a figure exemplifying the gating strategy is provided in the Supplementary Information.

