## [Peer Review File · Nature Genetics]

Peer Review Information

Manuscript Title: Mismatch repair deficiency is not sufficient to elicit tumor immunogenicity

Corresponding author name(s): Dr Tyler (E.) Jacks Dr Peter (Maxwell Kienitz) Dr Isidro Cortes-Ciriano Westcott

Editorial Notes:

Transferred manuscripts This document only contains reviewer comments, rebuttal and decision letters for versions considered at Nature Genetics.

Reviewer Comments & Decisions:

Decision Letter, initial version:

26th Apr 2023

Dear Dr Jacks,

Your Article, "Mismatch repair deficiency is not sufficient to elicit tumor immunogenicity" has now been seen by 2 referees. Reviewer #2 has a very minor request but Reviewer #1 continues to have some concerns about the alignment of your data with the model of evolution proposed. We are interested in the possibility of publishing your study in Nature Genetics, but would like to consider your response to these concerns in the form of a revised manuscript before we make a final decision on publication.

Reviewer #1 has proposed additional analyses to confirm accelerated neutral evolution in MMRd tumors. We agree that this would be an interesting analysis and you are very welcome to provide it. That said, considering the stage we are at, and the fact that this is a new request, the lack of these new data would not preclude our interest in the paper although we would encourage you to briefly address the point through textual edits. Their comment about multiple testing should be addressed in full.

Depending on how you choose to revise the paper, we'll either assess the revisions in-house, or we'll send the paper back out to one or more of your original reviewers. Please be assured that we are also reluctant to incur any further delays and will only do the latter if we deem it to be absolutely necessary.

We therefore invite you to revise your manuscript. Please highlight all changes in the manuscript text file. At this stage we will need you to upload a copy of the manuscript in MS Word .docx or similar editable format.

*2) If you have not done so already please begin to revise your manuscript so that it conforms to our Article format instructions, available [here](http://www.nature.com/ng/authors/article_types/index.html). Refer also to any guidelines provided in this letter.

[redacted]

We hope to receive your revised manuscript within four to eight weeks. If you cannot send it within this time, please let us know.

Sincerely,

Safia Danovi
Editor
Nature Genetics

Reviewers' Comments:

Reviewer #1:

Remarks to the Author:

Westcott and colleagues have further revised their study and addressed most of my concerns. I acknowledge that a lot of work has gone into this study, and there are interesting and relevant conclusions that I think are largely supported by the extensive data.

Specific comments:

* The exclusion of very low VAF mutations may be dampening the phenotype of the MMRd tumors (eg Fig 1e, Fig 3a-b). The authors explain this exclusion by referring to their pipeline, which takes the consensus of several standard somatic mutation calling algorithms. While PCAWG used a similar approach, they had a somewhat different goal, namely to build a high quality catalogue of clonal / oligo-clonal cancer driver mutations.

Here, the MMRd GEMM tumors likely have accelerated neutral somatic evolution, which should cause distinct patterns in the site frequency spectrum of somatic variants, particularly in the very low VAF regime. The shape of the CCF curve that is currently shown - particularly the peak - is unexpected. What is expected is a power law decay from 0 to 0.5 CCF. This was first demonstrated by Sottoriva and colleagues (Williams et al Nat Gen 2016 PMID 33571199, apologies to not include the citation before).

Their analysis, which was done on tumor exomes of similar depth as those in this study, showed that these very low VAF mutations are not sequencing noise but the result of neutral evolution. The shape of this decay allows you to directly estimate mutation rates (per base per division), which should be substantially higher in the KPM relative to KPC tumors.

Indeed studying these low VAF mutations requires somewhat modified variant calling approaches, not just the standard eg PCAWG style pipelines - Andrea's group for example has developed MOBSTER. It could also just be done by adjusting filters to include calls with low VAF and read count.

Such an analysis should demonstrate profound MMRd-specific elevation in the neutral mutation rates and subclonal mutation burdens. This should improve some key results that are now borderline, eg KPM vs KPC CCFs in Fig 3a. Again exomes at 100X should be well powered to detect elevated neutral mutation rates.

* The authors have clarified that many p values have already undergone multiple hypothesis correction. However, they also forego multiple hypothesis correction for analyses that involve two hypotheses eg 4d-e which have two hypotheses each.

I do not think this is appropriate, ie multiple hypothesis correction should be applied even for an analysis with two hypotheses. Also some analyses like 5e and 5f show only two hypotheses, but it is unclear a priori why the third hypothesis (0.5 100K vs 1.0 100K) would not be tested. Since most of these would be >0.05 by Bonferroni, I think they should be referred to as nominally significant.

Reviewer #2:

Remarks to the Author:

I am very grateful for their detailed response to my comments. The authors have done a thorough and excellent job on the revised manuscript. One of the referees comments that this is a negative study. However, from a clinical perspective, this referee considers the findings of critical oncology importance, highlighting the clinical caveats of inducing MMRd and subclonal neoantigens through drug therapy in patients to enhance an adaptive immune response which this referee and others have found problematic. The Jacks lab have now done a thorough job formally disproving this approach which will undoubtedly prevent patients coming to harm. This referee recommends enthusiastic publication and requests only that the authors include a discussion of this latest paper <https://pubmed.ncbi.nlm.nih.gov/36584674/>.

Author Rebuttal to Initial comments

Reviewers' Comments:

Reviewer #1:

Remarks to the Author:

Westcott and colleagues have further revised their study and addressed most of my concerns. I acknowledge that a lot of work has gone into this study, and there are interesting and relevant conclusions that I think are largely supported by the extensive data.

Specific comments:

* The exclusion of very low VAF mutations may be dampening the phenotype of the MMRd tumors (eg Fig 1e, Fig 3a-b). The authors explain this exclusion by referring to their pipeline, which takes the consensus of several standard somatic mutation calling algorithms. While PCAWG used a similar approach, they had a somewhat different goal, namely to build a high quality catalogue of clonal / oligo-clonal cancer driver mutations.

As we described in our previous point-by-point rebuttal, we have not excluded any mutations based on VAF. In addition, the minimum variant read threshold we employed (2 reads, the default for the callers used in our study: Mutect2, MuSE, Strelka2, VarDict) is lower than that used in the Sottoriva study (Williams, et al., *Nature Genetics*, 2016) referred to below, or 3 reads. We did not in fact take the consensus of all callers in our pipeline. Rather, as described in our Methods: "We only considered those SNVs mapping to exonic regions and detected by Mutect2 and supported by at least one of the other algorithms. To increase the accuracy of indel detection, only indels detected by at least 2 algorithms were considered for further analysis." In this way, we take advantage of the higher sensitivity afforded by Mutect2 while also guaranteeing high specificity. We stand by this pipeline and believe it is an optimal balance between permissive calling and exclusion of noise/erroneous calls. We do acknowledge that the pipeline in the Sottoriva study used only one caller, which is certainly subject to more caller-specific error. Regardless of the goal of PCAWG or our study, we think it is important to take some measure to reduce this type of error.

Here, the MMRd GEMM tumors likely have accelerated neutral somatic evolution, which should cause distinct patterns in the site frequency spectrum of somatic variants, particularly in the very low VAF regime. The shape of the CCF curve that is currently shown - particularly the peak - is unexpected. What is expected is a power law decay from

0 to 0.5 CCF. This was first demonstrated by Sottoriva and colleagues (Williams et al Nat Gen 2016 PMID 33571199, apologies to not include the citation before).

Directly comparing CCF versus VAF distributions is fraught, as these metrics, while fundamentally related, are calculated differently. VAF is simply variant over total reads, while CCF is VAF corrected for tumor purity and ploidy and, if computed correctly, is a superior metric of clonality. Given that tumors in our study were not 100% pure and genome doubling was frequent, it is expected that our CCF distributions appear right-shifted compared to VAFs in the Sottoriva study. In Fig. 1a of the Sottoriva study, the VAF peak is at ~0.06. Adjusting for an average purity (in our study) of ~0.75 and genome doubling (1 variant to 3 wt reads at clonality), the corresponding CCF peak is 0.16, not so dissimilar from the CCF peaks in our study.

Their analysis, which was done on tumor exomes of similar depth as those in this study, showed that these very low VAF mutations are not sequencing noise but the result of neutral evolution. The shape of this decay allows you to directly estimate mutation rates (per base per division), which should be substantially higher in the KPM relative to KPC tumors.

Indeed studying these low VAF mutations requires somewhat modified variant calling approaches, not just the standard eg PCAWG style pipelines - Andrea's group for example has developed MOBSTER. It could also just be done by adjusting filters to include calls with low VAF and read count.

Again, our variant calling approach, while more stringent from the perspective of requiring agreement between at least two of four independent callers (one of which is Mutect2, known to have high sensitivity), actually employed a lower variant allele threshold of 2 reads. To definitively show that this referee's observations are not due to differences in calling pipelines but rather the differences in distributions between VAF and CCF, we include here VAF distributions and neutral mutation rates in our study, which are entirely consistent with those in the Sottoriva study:

VAF:

Purity corrected VAF:

We thank this referee for raising the point about neutral mutation evolution and the theoretical framework to measure it. We now include these cumulative distribution plots of subclonal mutations ($M(f)$) for lung and colon $Msh2^{KO}$ tumors from our models in Fig. 3c-d, following the formulas described in the Sottoriva study, demonstrating clearly that mutations in our models are shaped by neutral evolutionary dynamics. This is discussed in the text at lines 193-195:

“These mutations also adhered perfectly to a theoretical model of neutral evolution of subclonal mutations in cancer⁴³ (Fig. 3c-d), consistent with the absence of selective events following tumor initiation in our models.”

We also now raise this point in the Discussion (lines 496-499):

“... our models, which more closely resemble sporadic MMRd, followed a model of neutral evolutionary dynamics⁴³ and did not display increased baseline immunogenicity or response to ICB, likely due to timing of MMR inactivation and resulting patterns of clonal expansion⁵².”

Such an analysis should demonstrate profound MMRd-specific elevation in the neutral mutation rates and subclonal mutation burdens. This should improve some key results that are now borderline, eg KPM vs KPC CCFs in Fig 3a. Again exomes at 100X should be well powered to detect elevated neutral mutation rates.

We appreciate this referee’s good-faith effort to assist us in leveraging the full extent of subclonal mutations in our data, but we are confident that we have maximized signal-to-noise in our mutation calling and as we show above, see highly comparable levels of neutrally accumulated mutations as in the Sottoriva study. The tumors in our mouse models developed over a shorter period of time and lack major selective events after initiation, therefore representing an even more extreme degree of neutral evolution. While we agree that the MMRd tumors in our models demonstrate profound elevation in neutral/subclonal mutation rates, as we showed in our single cell cloning and sequencing analyses, 100X WES is simply not sufficient to capture the vast majority of these subclonal events, especially in the context of early genome doubling.

* The authors have clarified that many p values have already undergone multiple hypothesis correction. However, they also forego multiple hypothesis correction for analyses that involve two hypotheses eg 4d-e which have two hypotheses each.

I do not think this is appropriate, ie multiple hypothesis correction should be applied even for an analysis with two hypotheses. Also some analyses like 5e and 5f show only two hypotheses, but it is unclear a priori why the third hypothesis (0.5 100K vs 1.0 100K) would not be tested. Since most of these would be >0.05 by Bonferroni, I think they should be referred to as nominally significant.

We appreciate this referee's focus on statistical rigor. Throughout our manuscript we performed only non-parametric tests, which have fewer assumptions but are less powered than tests like the standard t-test (by far the most commonly employed even in the absence of tests of normality). We also performed multiple test correction on all figure panels with more than two tests, which is extremely rigorous compared to standards in the field, especially considering that many of these panels test independent hypotheses. To provide context, a study published in *Nature Genetics* on April 24, 2023 (Edahiro, R, et al., 2023, *Nature Genetics*) did not perform multiple test correction for most figures, including Fig. 3d, which has 8 tests. Transparently, the authors refer to these p values in the figure legend as $P_{\text{Uncorrected}}$. For those panels in our manuscript that have only two tests of independent hypotheses, we now refer to the p values in the figure legend as $P_{\text{Uncorrected}}$, like Edahiro, et al. For all other instances we perform multiple test correction.

We have also made some additional changes that we are confident would satisfy this referee. Regarding Fig. 5e-f, the referee states "it is unclear a priori why the third hypothesis (0.5 100K vs 1.0 100K) would not be tested". We would first note that in a well-designed study with a priori hypotheses, it is not necessary nor good statistical practice to perform every statistical comparison possible. Regarding the "0.5 100k vs 1.0 100k" comparison, we agree this comparison is interesting, but it is also a distinctly different hypothesis from the main hypothesis of Fig. 5, that clonality correlates with magnitude and effector differentiation of the T cell response. "0.5 100k vs 1.0 100k" asks whether it is more so clonality or the total number of neoantigen-expressing tumor cells that is the dominant driver of the observed effects. Therefore, we now present separate panels (Fig. 5d,g,i) where we make the "0.5 100k vs 1.0 100k" comparison. We also now better leverage the clear trends in Fig. 5c,f,h (previously Fig. 5c,e,f), which are more suited to statistical tests of correlation rather than pairwise comparison, presenting significant Spearman Correlations in addition to a single Wilcoxon Rank Sum (non-parametric) comparison of the most extreme groups (CCF 1.0 versus 0.125):

Figure 5

Finally, in Fig. 4d-e we now show p values calculated from t-test, not Wilcoxon Rank Sum, as this referee flagged these plots as borderline significant and we therefore felt it important to reevaluate the statistical test employed. We verified that these data are roughly normally distributed (Q-Q plot visualization and Shapiro-Wilk normality test $p > 0.05$). Therefore, t-test is a more appropriate and powered test of significance:

Importantly, all of these minor changes to statistical tests and reported p values have had no impact on interpretation or presentation of the results in the manuscript.

Reviewer #2:
Remarks to the Author:

I am very grateful for their detailed response to my comments. The authors have done a thorough and excellent job on the revised manuscript. One of the referees comments that this is a negative study. However, from a clinical perspective, this referee considers the findings of critical oncology importance, highlighting the clinical caveats of inducing MMRd and subclonal neoantigens through drug therapy in patients to enhance an adaptive immune response which this referee and others have found problematic. The Jacks lab have now done a thorough job formally disproving this approach which will undoubtedly prevent patients coming to harm. This referee recommends enthusiastic publication and requests only that the authors include a discussion of this latest paper <https://pubmed.ncbi.nlm.nih.gov/36584674/>.

We are thankful for this referee's thoughtful comments and support throughout the review process, which have greatly improved the manuscript and its potential impact.

We now include a discussion of the aforementioned paper (lines 516-520):

"Recently, a preclinical study showed that genetic or pharmacological enrichment of MMRd in the context of mixed MMRd/MMRp cell line transplants potentiated rejection of the MMRp fraction⁵⁴. This must be interpreted with care, however, as both MMRd and MMRp fractions were derived from the same carcinogen (N-nitroso-N-methylurethane)-induced colon carcinoma line, CT26, and likely share a high burden of clonal neoantigens that may underlie rejection of the MMRp fraction."

Decision Letter, first revision:

22nd May 2023

Dear Drs Jacks and Westcott,

Thank you so much for bearing with me while I completed the checks on your paper. I am grateful for your patience.

Thank you for submitting your revised manuscript "Mismatch repair deficiency is not sufficient to elicit tumor immunogenicity" (NG-A62150R). We assessed your revisions in-house and I'm delighted to say

that we'll be happy in principle to publish it in Nature Genetics, pending minor revisions to satisfy our editorial and formatting guidelines.

Sincerely,

Safia

Safia Danovi
Editor
Nature Genetics

Final Decision Letter:

Pending ACC